# Online Planning with Lookahead Policies

Yonathan Efroni [*][†][‡]       Mohammad Ghavamzadeh [§]       Shie Mannor [‡][¶]

## Abstract

Real Time Dynamic Programming (RTDP) is an online algorithm based on Dynamic Programming (DP) that acts by 1-step greedy planning. Unlike DP, RTDP does not require access to the entire state space, i.e., it explicitly handles the exploration. This fact makes RTDP particularly appealing when the state space is large and it is not possible to update all states simultaneously. In this we devise a multi-step greedy RTDP algorithm, which we call $h$-RTDP, that replaces the 1-step greedy policy with a $h$-step lookahead policy. We analyze $h$-RTDP in its exact form and establish that increasing the lookahead horizon, $h$, results in an improved sample complexity, with the cost of additional computations. This is the first work that proves improved sample complexity as a result of *increasing* the lookahead horizon in online planning. We then analyze the performance of $h$-RTDP in three approximate settings: approximate model, approximate value updates, and approximate state representation. For these cases, we prove that the asymptotic performance of $h$-RTDP remains the same as that of a corresponding approximate DP algorithm, the best one can hope for without further assumptions on the approximation errors.

## 1 Introduction

Dynamic Programming (DP) algorithms return an optimal policy, given a model of the environment. Their convergence in the presence of lookahead policies [4, 13] and their performance in different approximate settings [4, 25, 27, 17, 1, 14] have been well-studied. Standard DP algorithms require simultaneous access to the *entire* state space at run time, and as such, cannot be used in practice when the number of states is too large. Real Time Dynamic Programming (RTDP) [3, 29] is a DP-based algorithm that mitigates the need to access all states simultaneously. Similarly to DP, RTDP updates are based on the Bellman operator, calculated by accessing the model of the environment. However, unlike DP, RTDP learns how to act by interacting with the environment. In each episode, RTDP interacts with the environment, acts according to the greedy action w.r.t. the Bellman operator, and samples a trajectory. RTDP is, therefore, an online planning algorithm.

Despite the popularity and simplicity of RTDP and its extensions [5, 6, 24, 8, 29, 22], precise characterization of its convergence was only recently established for finite-horizon MDPs [15]. While lookahead policies in RTDP are expected to improve the convergence in some of these scenarios, as they do for DP [4, 13], to the best of our knowledge, these questions have not been addressed in previous literature. Moreover, previous research haven't addressed the questions of how lookahead policies should be used in RTDP, nor studied RTDP's sensitivity to possible approximation errors. Such errors can arise due to a misspecified model, or exist in value function updates, when e.g., function approximation is used.

---

[*]Part of this work was done during an internship in Facebook AI Research

[†]Microsoft Research, New York, NY

[‡]Technion, Israel

[§]Google Research

[¶]Nvidia Research

In this paper, we initiate a comprehensive study of lookahead-policy based RTDP with approximation errors in *online planning*. We start by addressing the computational complexity of calculating lookahead policies and study its advantages in approximate settings. Lookahead policies can be computed naively by exhaustive search in $O(A^h)$ for deterministic environments or $O(A^{Sh})$ for stochastic environments. Since such an approach is infeasible, we offer in Section 3 an alternative approach for obtaining a lookahead policy with a computational cost that depends linearly on a natural measure: the total number of states reachable from a state in $h$ time steps. The suggested approach is applicable both in deterministic and stochastic environments.

In Section 5, we introduce and analyze $h$-RTDP, a RTDP-based algorithm that replaces the 1-step greedy used in RTDP by a $h$-step lookahead policy. The analysis of $h$-RTDP reveals that the sample complexity is improved by increasing the lookahead horizon $h$. To the best of our knowledge, this is the first theoretical result that relates sample complexity to the lookahead horizon in online planning setting. In Section 6, we analyze $h$-RTDP in the presence of three types of approximation: when (i) an inexact model is used, instead of the true one, (ii) the value updates contain error, and finally (iii) approximate state abstraction is used. Interestingly, for approximate state abstraction, $h$-RTDP convergence and computational complexity depends on the size of the *abstract state space*.

In a broader context, this work shows that RTDP-like algorithms could be a good alternative to Monte Carlo tree search (MCTS) [7] algorithms, such as upper confidence trees (UCT) [21], an issue that was empirically investigated in [22]. We establish strong convergence guarantees for extensions of $h$-RTDP: under no assumption other than initial optimistic value, RTDP-like algorithms combined with lookahead policies converge in polynomial time to an optimal policy (see Table 1), and their approximations inherit the asymptotic performance of approximate DP (ADP). Unlike RTDP, MCTS acts by using a $\sqrt{\log N/N}$ bonus term instead of optimistic initialization. However, in general, its convergence can be quite poor, even worse than uniformly random sampling [9, 26].

## 2  Preliminaries

**Finite Horizon MDPs.**  A finite-horizon MDP [4] with time-independent dynamics[6] is a tuple $\mathcal{M} = (\mathcal{S}, \mathcal{A}, r, p, H)$, where $\mathcal{S}$ and $\mathcal{A}$ are the state and action spaces with cardinalities $S$ and $A$, respectively, $r(s, a) \in [0, 1]$ is the immediate reward of taking action $a$ at state $s$, and $p(s'|s, a)$ is the probability of transitioning to state $s'$ upon taking action $a$ at state $s$. The initial state in each episode is arbitrarily chosen and $H \in \mathbb{N}$ is the MDP's horizon. For any $N \in \mathbb{N}$, denote $[N] := \{1, \ldots, N\}$.

A deterministic policy $\pi : \mathcal{S} \times [H] \rightarrow \mathcal{A}$ is a mapping from states and time step indices to actions. We denote by $a_t := \pi_t(s)$ the action taken at time $t$ at state $s$ according to a policy $\pi$. The quality of a policy $\pi$ from a state $s$ at time $t$ is measured by its value function, i.e., $V_t^\pi(s) := \mathbb{E}\big[ \sum_{t'=t}^{H} r(s_{t'}, \pi_{t'}(s_{t'})) \mid s_t = s \big]$, where the expectation is over all the randomness in the environment. An optimal policy maximizes this value for all states $s \in \mathcal{S}$ and time steps $t \in [H]$, i.e., $V_t^*(s) := \max_\pi V_t^\pi(s)$, and satisfies the optimal Bellman equation,

$$V_t^*(s) = TV_{t+1}^*(s) := \max_a \big( r(s, a) + p(\cdot|s, a)V_{t+1}^* \big)$$
$$= \max_a \mathbb{E}\big[ r(s_1, a) + V_{t+1}^*(s_2) \mid s_1 = s \big]. \tag{1}$$

By repeatedly applying the optimal Bellman operator $T$, for any $h \in [H]$, we have

$$V_t^*(s) = T^h V_{t+h}^*(s) = \max_a \big( r(s, a) + p(\cdot|s, a)T^{h-1}V_{t+h}^* \big)$$

$$= \max_{\pi_t, \ldots, \pi_{t+h-1}} \mathbb{E}\Big[ \sum_{t'=1}^{h} r(s_{t'}, \pi_{t+t'-1}(s_{t'})) + V_{t+h}^*(s_{h+1}) \mid s_1 = s \Big]. \tag{2}$$

We refer to $T^h$ as the $h$-step optimal Bellman operator. Similar Bellman recursion is defined for the value of a given policy, $\pi$, i.e., $V^\pi$, as $V_t^\pi(s) = T_\pi^h V_{t+h}^\pi(s) := r(s, \pi_t(s)) + p(\cdot|s, \pi_t(s))T_\pi^{h-1}V_{t+h}^\pi$, where $T_\pi^h$ is the $h$-step Bellman operator of policy $\pi$.

$h$-**Lookahead Policy.** An $h$-lookahead policy w.r.t. a value function $V \in \mathbb{R}^S$ returns the optimal first action in an $h$-horizon MDP. For a state $s \in \mathcal{S}$, it returns

$$a_h(s) \in \arg\max_a \big(r(s,a) + p(\cdot|s,a)T^{h-1}V\big)$$

$$= \arg\max_{\pi_1(s)} \max_{\pi_2,\dots,\pi_h} \mathbb{E}\Big[\sum_{t=1}^h r(s_t, \pi_t(s_t)) + V(s_{h+1})|s_1 = s\Big]. \tag{3}$$

We can see $V$ represent our 'prior-knowledge' of the problem. For example, it is possible to show [4] that if $V$ is close to $V^*$, then the value of a $h$-lookahead policy w.r.t. $V$ is close to $V^*$.

For a state $s \in \mathcal{S}$ and a number of time steps $h \in [H]$, we define the set of reachable states from $s$ in $h$ steps as $\mathcal{S}_h(s) = \{s' \mid \exists \pi : p^\pi(s_{h+1} = s' \mid s_1 = s, \pi) > 0\}$, and denote by $S_h(s)$ its cardinality. We define the set of reachable states from $s$ in up to $h$ steps as $\mathcal{S}_h^{Tot}(s) := \cup_{t=1}^h \mathcal{S}_t(s)$, its cardinality as $S_h^{Tot}(s) := \sum_{t=1}^h S_t(s)$, and the maximum of this quantity over the entire state space as $S_h^{Tot} = \max_s S_h^{Tot}(s)$. Finally, we denote by $\mathcal{N} := S_1^{Tot}$ the maximum number of accessible states in 1-step (neighbors) from any state.

**Regret and Uniform-PAC.** We consider an agent that repeatedly interacts with an MDP in a sequence of episodes $[K]$. We denote by $s_t^k$ and $a_t^k$, the state and action taken at the time step $t$ of the $k$'th episode. We denote by $\mathcal{F}_{k-1}$, the filtration that includes all the events (states, actions, and rewards) until the end of the $(k-1)$'th episode, as well as the initial state of the $k$'th episode. Throughout the paper, we denote by $\pi_k$ the policy that is executed during the $k$'th episode and assume it is $\mathcal{F}_{k-1}$ measurable. The performance of an agent is measured by its *regret*, defined as $\text{Regret}(K) := \sum_{k=1}^K \big(V_1^*(s_1^k) - V_1^{\pi_k}(s_1^k)\big)$, as well as by the *Uniform-PAC* criterion [10], which we generalize to deal with approximate convergence. Let $\epsilon, \delta > 0$ and $N_\epsilon = \sum_{k=1}^\infty \mathbb{1}\{V_1^*(s_1^k) - V_1^{\pi_k}(s_1^k) \geq \epsilon\}$ be the number of episodes in which the algorithm outputs a policy whose value is $\epsilon$-inferior to the optimal value. An algorithm is called Uniform-PAC, if $\Pr(\exists \epsilon > 0 : N_\epsilon \geq F(S, 1/\epsilon, \log 1/\delta, H)) \leq \delta$, where $F(\cdot)$ depends polynomially (at most) on its parameters. Note that Uniform-PAC implies $(\epsilon, \delta)$-PAC, and thus, it is a stronger property. As we analyze algorithms with inherent errors in this paper, we use a more general notion of $\Delta$-Uniform-PAC by defining the random variable $N_\epsilon^\Delta = \sum_{k=1}^\infty \mathbb{1}\{V_1^*(s_1^k) - V_1^{\pi_k}(s_1^k) \geq \Delta + \epsilon\}$, where $\Delta > 0$. Finally, we use $\tilde{\mathcal{O}}(x)$ to represent $x$ up to constants and poly-logarithmic factors in $\delta$, and $O(x)$ to represent $x$ up to constants.

## 3 Computing $h$-Lookahead Policies

Computing an action returned by a $h$-lookahead policy at a certain state is a main component in the RTDP-based algorithms we analyze in Sections 5 and 6. A 'naive' procedure that returns such action is the exhaustive search. Its computational cost is $O(A^h)$ and $O(A^{Sh})$ for deterministic and stochastic systems, respectively. Such an approach is impractical, even for moderate values of $h$ or $S$.

Instead of the naive approach, we formulate a Forward-Backward DP (FB-DP) algorithm, whose pseudo-code is given in Appendix 9. The FB-DP returns an action of an $h$-lookahead policy from a given state $s$. Importantly, in both deterministic and stochastic systems, the computation cost of FB-DP depends linearly on the total *number of reachable states* from $s$ in up to $h$ steps, i.e., $S_h^{Tot}(s)$. In the worst case, we may have $S_h(s) = O(\min(A^h, S))$. However, when $S_h^{Tot}(s)$ is small, significant improvement is achieved by avoiding unnecessary repeated computations.

FB-DP has two subroutines. It first constructs the set of reachable states from state $s$ in up to $h$ steps, $\{\mathcal{S}_t(s)\}_{t=1}^h$, in the 'forward-pass'. Given this set, in the second 'backward-pass' it simply applies backward induction (Eq. 3) and returns an action suggested by the $h$-lookahead policy, $a_h(s)$. Note that at each stage $t \in [h]$ of the backward induction (applied on the set $\{\mathcal{S}_t(s)\}_{t=1}^h$) there are $S_t(s)$ states on which the Bellman operator is applied. Since applying the Bellman operator costs $O(\mathcal{N}A)$ computations, the computational cost of the 'backward-pass' is $O(\mathcal{N}AS_h^{Tot}(s))$.

In Appendix 9, we describe a DP-based approach to efficiently implement 'forward-pass' and analyze its complexity. Specifically, we show the computational cost of the 'forward-pass' is equivalent to that of the 'backward-pass' (see Propsition 8). Meaning, the computational cost of FB-DP is $O(\mathcal{N}AS_h^{Tot}(s))$ - same order as the cost of backward induction given the set $\mathcal{S}_h^{Tot}(s)$.

# 4 Real-Time Dynamic Programming

Real-time dynamic programming (RTDP) [3] is a well-known online planning algorithm that assumes access to a transition model and a reward function. Unlike DP algorithms (policy, value iteration, or asynchronous value iteration) [4] that solve an MDP using offline calculations and sweeps over the entire states (possibly in random order), RTDP solves it in real-time, using samples from the environment (either simulated or real) and DP-style Bellman updates from the current state. Furthermore, unlike DP algorithms, RTDP needs to tradeoff exploration-exploitaion, since it interacts with the environment via sampling trajectories. This makes RTDP a good candidate for problems in which having access to the entire state space is not possible, but interaction is.

Algorithm 1 contains the pseudo-code of RTDP in finite-horizon MDPs. The value is initialized optimistically, $\bar{V}_{t+1}^0(s) = H - t \geq V_{t+1}^*(s)$. At each time step $t \in [H]$ and episode $k \in [K]$, the agent updates the value of the current state $s_t^k$ by the optimal Bellman operator. It then acts greedily w.r.t. the current value at the next time step $\bar{V}_{t+1}^{k-1}$. Finally, the next state, $s_{t+1}^k$, is sampled either from the model or the real-world. When the model is exact, there is no difference in sampling from the model and real-world, but these are different in case the model is inexact as in Section 6.1.

The following high probability bound on the regret of a Decreasing Bounded Process (DBP), proved in [15], plays a key role in our analysis of exact and approximate RTDP with lookahead policies in Sections 5 and 6. An adapted process $\{X_k, \mathcal{F}_k\}_{k \geq 0}$ is a DBP, if for all $k \geq 0$, (i) $X_k \leq X_{k-1}$ almost surely (a.s.), (ii) $X_k \geq C_2$, and (iii) $X_0 = C_1 \geq C_2$. Interestingly, contrary to the standard regret bounds (e.g., in bandits), this bound does not depend on the number of rounds $K$.

**Theorem 1** (Regret Bound of a DBP [15]). *Let $\{X_k, \mathcal{F}_k\}_{k \geq 0}$ be a DBP and $R_K = \sum_{k=1}^K X_{k-1} - \mathbb{E}[X_k \mid \mathcal{F}_{k-1}]$ be its $K$-round regret. Then,*

$$\Pr\{\exists K > 0 : R_K \geq 9(C_1 - C_2)\ln(3/\delta)\} \leq \delta.$$

# 5 RTDP with Lookahead Policies

In this section, we devise and analyze a lookahead-based RTDP algorithm, called $h$-RTDP, whose pseudo-code is shown in Algorithm 2. Without loss of generality, we assume that $H/h \in \mathbb{N}$. We divide the horizon $H$ into $H/h$ intervals, each of length $h$ time steps. $h$-RTDP stores $HS/h$ values in the memory, i.e., the values at time steps $\mathcal{H} = \{1, h+1, \ldots, H+1\}$.[7] For each time step $t \in [H]$, we denote by $h_c \in \mathcal{H}$, the next time step for which a value is stored in the memory, and by $t_c = h_c - t$, the number of time steps until there (see Figure 1). At each time step $t$ of an episode $k \in [K]$, given the current state $s_t^k$, $h$-RTDP selects an action $a_t^k$ returned by the $t_c$-lookahead policy w.r.t. $\bar{V}_{h_c}^{k-1}$,

$$a_t^k = a_{t_c}(s_t^k) \in \arg\max_{\pi_1(s_t^k)} \max_{\pi_2, \ldots, \pi_{t_c}} \mathbb{E}\left[ \sum_{t'=1}^{t_c} r(s_{t'}, \pi_{t'}(s_{t'})) + \bar{V}_{h_c}^{k-1}(s_{t_c+1}) \mid s_1 = s_t^k \right]. \quad (4)$$

Thus, $h$-RTDP uses a varying lookahead horizon $t_c$ that depends on how far the current time step is to the next one for which a value is stored. Throughout the paper, with an abuse of notation, we refer to this policy as a $h$-lookahead policy. Finally, it can be seen that $h$-RTDP generalizes RTDP as they are equal for $h = 1$.

We are now ready to establish finite-sample performance guarantees for $h$-RTDP; see Appendix 10 for the detailed proofs. We start with two lemmas from which we derive the main convergence result of this section.

**Lemma 2.** *For all $s \in \mathcal{S}$, $n \in \{0\} \cup [\frac{H}{h}]$, and $k \in [K]$, the value function of $h$-RTDP is (i) Optimistic: $V_{nh+1}^*(s) \leq \bar{V}_{nh+1}^k(s)$, and (ii) Non-Increasing: $\bar{V}_{nh+1}^k(s) \leq \bar{V}_{nh+1}^{k-1}(s)$.*

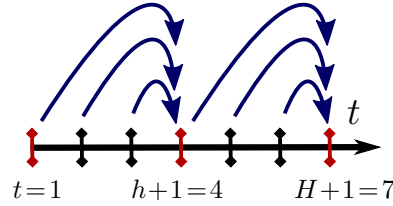

Figure 1: Varying lookahead horizon of a $h$-greedy policy in $h$-RTDP (see Eq. 4) with $h = 3$ and $H = 6$. The blue arrows show the lookahead horizon from a specific time step $t$, and the red bars are the time steps for which a value is stored in memory, i.e., $\mathcal{H} = \{1, h+1 = 4, 2h+1 = H+1 = 7\}$.

| **Algorithm 1** Real-Time DP (RTDP) | **Algorithm 2** RTDP with Lookahead ($h$-RTDP) |
|---|---|

**Algorithm 1** Real-Time DP (RTDP)

**init:** $\forall s \in \mathcal{S}, \ \forall t \in \{0\} \cup [H]$,
$\quad \bar{V}_{t+1}^0(s) = H - t$
**for** $k \in [K]$ **do**
$\quad$ Initialize $s_1^k$ arbitrarily
$\quad$ **for** $t \in [H]$ **do**
$\quad\quad \bar{V}_t^k(s_t^k) = T\bar{V}_{t+1}^{k-1}(s_t^k)$
$\quad\quad a_t^k \in \arg\max_a r(s_t^k, a) + p(\cdot|s_t^k, a)\bar{V}_{t+1}^{k-1}$
$\quad\quad$ Act by $a_t^k$, observe $s_{t+1}^k \sim p(\cdot \mid s_t^k, a_t^k)$
$\quad$ **end for**
**end for**

**Algorithm 2** RTDP with Lookahead ($h$-RTDP)

**init::** $\forall s \in \mathcal{S}, \ n \in \{0\} \cup [\frac{H}{h}]$,
$\quad \bar{V}_{nh+1}^0(s) = H - nh$
**for** $k \in [K]$ **do**
$\quad$ Initialize $s_1^k$ arbitrarily
$\quad$ **for** $t \in [H]$ **do**
$\quad\quad$ **if** $(t-1) \mod h = 0$ **then**
$\quad\quad\quad h_c = t + h$
$\quad\quad\quad \bar{V}_t^k(s_t^k) = T^h \bar{V}_{h_c}^{k-1}(s_t^k)$
$\quad\quad$ **end if**
$\quad\quad a_t^k \in$
$\quad\quad \arg\max_a r(s_t^k, a) + p(\cdot|s_t^k, a)T^{h_c-t-1}\bar{V}_{h_c}^{k-1}$
$\quad\quad$ Act by $a_t^k$, observe $s_{t+1}^k \sim p(\cdot \mid s_t^k, a_t^k)$
$\quad$ **end for**
**end for**

**Lemma 3** (Optimality Gap and Expected Decrease). *The expected cumulative value update at the $k$'th episode of $h$-RTDP satisfies* $\bar{V}_1^k(s_1^k) - V_1^{\pi_k}(s_1^k) = \sum_{n=1}^{\frac{H}{h}-1} \sum_{s \in \mathcal{S}} \bar{V}_{nh+1}^{k-1}(s) - \mathbb{E}[\bar{V}_{nh+1}^k(s) \mid \mathcal{F}_{k-1}]$.

Properties (i) and (ii) in Lemma 2 show that $\{\bar{V}_{nh+1}^k(s)\}_{k \geq 0}$ is a DBP, for any $s$ and $n$. Lemma 3 relates $\bar{V}_1^k(s_1^k) - V_1^{\pi_k}(s_1^k)$ (LHS) to the expected decrease in $\bar{V}^k$ at the $k$'th episode (RHS). When the LHS is small, then $\bar{V}_1^k(s_1^k) \simeq V_1^*(s_1^k)$, due to the optimism of $\bar{V}_1^k$, and $h$-RTDP is about to converge to the optimal value. This is why we refer to the LHS as the *optimality gap*. Using these two lemmas and the regret bound of a DBP (Theorem 1), we prove a finite-sample convergence result for $h$-RTDP (see Appendix 10 for the full proof).

**Theorem 4** (Performance of $h$-RTDP). *Let $\epsilon, \delta > 0$. The following holds for $h$-RTDP:*

1. *With probability* $1 - \delta$, *for all* $K > 0$, $\quad \text{Regret}(K) \leq \frac{9SH(H-h)}{h} \ln(3/\delta)$.

2. $\Pr\left\{ \exists \epsilon > 0 \ : \ N_\epsilon \geq \frac{9SH(H-h)\ln(3/\delta)}{h\epsilon} \right\} \leq \delta$.

*Proof Sketch.* Applying Lemmas 2 and 3, we may write

$$\text{Regret}(K) \leq \sum_{k=1}^K \bar{V}_1^k(s_1^k) - V_1^{\pi_k}(s_1^k) = \sum_{k=1}^K \sum_{n=1}^{\frac{H}{h}-1} \sum_s \bar{V}_{nh+1}^{k-1}(s) - \mathbb{E}[\bar{V}_{nh+1}^k(s) \mid \mathcal{F}_{k-1}]$$

$$= \sum_{k=1}^K X_{k-1} - \mathbb{E}[X_k \mid \mathcal{F}_{k-1}]. \tag{5}$$

Where we define $X_k := \sum_{n=1}^{\frac{H}{h}-1} \sum_s \bar{V}_{nh+1}^{k-1}(s)$ and use linearity of expectation. By Lemma 2, $\{X_k\}_{k \geq 0}$ is decreasing and bounded from below by $\sum_{n=1}^{\frac{H}{h}-1} \sum_s V_{nh+1}^*(s) \geq 0$. We conclude the proof by observing that $X_0 \leq \sum_{n=1}^{\frac{H}{h}-1} \sum_s V_{nh+1}^0(s) \leq SH(H-h)/h$, and applying Theorem 1. $\quad \square$

**Remark 1** (RTDP and Good Value Initialization). *A closer look into the proof of Theorem 4 shows we can easily obtain a stronger result which depends on the initial value $V^0$. The regret can be bounded by* $\text{Regret}(K) \leq \tilde{\mathcal{O}}\left( \sum_{n=1}^{\frac{H}{h}-1} \left( V_{nh+1}^0(s) - V_{nh+1}^*(s) \right) \right)$, *which formalizes the intuition the algorithm improves as the initial value $V^0$ better estimates $V^*$. For clarity purposes we provide the worse-case bound.*

**Remark 2** (Computational Complexity of $h$-RTDP). *Using FB-DP (Section 3) as a solver of a $h$-lookahead policy, the per-episode computation cost of $h$-RTDP amounts to applying FB-DP for $H$ time steps, i.e., it is bounded by $O(H\mathcal{N}AS_h^{Tot})$. Since $S_h^{Tot}$ – the total number of reachable states in up to $h$ time steps – is an increasing function of $h$, the computation cost of $h$-RTDP increases with $h$, as expected. When $S_h^{Tot}$ is significantly smaller than $S$, the per-episode computational complexity of*

$h$-*RTDP is $S$ independent. As discussed in Section 3, using FB-DP, in place of exhaustive search, can significantly improve the computational cost of $h$-RTDP.*

**Remark 3** (Improved Sample Complexity of $h$-RTDP). *Theorem 4 shows that $h$-RTDP improves the sample complexity of RTDP by a factor $1/h$. This is consistent with the intuition that larger horizon of the applied lookahead policy results in faster convergence (less samples). Thus, if RTDP is used in a real-time manner, one way to boost its performance is to combine with lookahead policies.*

**Remark 4** (Sparse Sampling Approaches). *In this work, we assume $h$-RTDP has access to a $h$-lookahead policy (3) solver, such as FB-DP presented in Section 3. We leave studying the sparse sampling approach [19, 28] for approximately solving $h$-lookahead policy for future work.*

## 6 Approximate RTDP with Lookahead Policies

In this section, we consider three approximate versions of $h$-RTDP in which the update deviates from its exact form described in Section 5. We consider the cases in which there are errors in the **1)** *model*, **2)** *value updates*, and when we use **3)** *approximate state abstraction*. We prove finite-sample bounds on the performance of $h$-RTDP in the presence of these approximations. Furthermore, in Section 6.3, given access to an approximate state abstraction, we show that the convergence of $h$-RTDP depends on the cardinality of the *abstract state space* – which can be much smaller than the original one. The proofs of this section generalize that of Theorem 4, while following the same 'recipe'. This shows the generality of the proof technique, as it works for both exact and approximate settings.

### 6.1 $h$-**RTDP with Approximate Model ($h$-RTDP-AM)**

In this section, we analyze a more practical scenario in which the transition model used by $h$-RTDP to act and update the values is not exact. We assume it is close to the true model in the total variation $(TV)$ norm, $\forall (s,a) \in \mathcal{S} \times \mathcal{A}$, $||p(\cdot|s,a) - \hat{p}(\cdot|s,a)||_1 \leq \epsilon_P$, where $\hat{p}$ denotes the approximate model. Throughout this section and the relevant appendix (Appendix 11), we denote by $\hat{T}$ and $\hat{V}^*$ the optimal Bellman operator and optimal value of the approximate model $\hat{p}$, respectively. Note that $\hat{T}$ and $\hat{V}^*$ satisfy (1) and (2) with $p$ replaced by $\hat{p}$. $h$-RTDP-AM is exactly the same as $h$-RTDP (Algorithm 2) with the model $p$ and optimal Bellman operator $T$ replaced by their approximations $\hat{p}$ and $\hat{T}$. We report the pseudocode of $h$-RTDP-AM in Appendix 11.

Although we are given an approximate model, $\hat{p}$, we are still interested in the performance of (approximate) $h$-RTDP on the *true MDP*, $p$, and relative to its optimal value, $V^*$. If we solve the approximate model and act by its optimal policy, the Simulation Lemma [20, 30] suggests that the regret is bounded by $O(H^2\epsilon_P K)$. For $h$-RTDP-AM, the situation is more involved, as its updates are based on the approximate model and the samples are gathered by interacting with the true MDP. Nevertheless, by properly adjusting the techniques from Section 5, we derive performance bounds for $h$-RTDP-AM. These bounds reveal that the asymptotic regret increases by at most $O(H^2\epsilon_P K)$, similarly to the regret of the optimal policy of the approximate model. Interestingly, the proof technique follows that of the exact case in Theorem 4. We generalize Lemmas 2 and 3 from Section 5 to the case that the update rule uses an inexact model (see Lemmas 9 and 10 in Appendix 11). This allows us to establish the following performance bound for $h$-RTDP-AM (proof in Appendix 11).

**Theorem 5** (Performance of $h$-RTDP-AM). *Let $\epsilon, \delta > 0$. The following holds for $h$-RTDP-AM:*

1. *With probability $1 - \delta$, for all $K > 0$,* $\quad \mathrm{Regret}(K) \leq \frac{9SH(H-h)}{h}\ln(3/\delta) + H(H-1)\epsilon_P K$.

2. *Let $\Delta_P = H(H-1)\epsilon_P$. Then,* $\quad \mathrm{Pr}\left\{\exists \epsilon > 0 \ : \ N_\epsilon^{\Delta_P} \geq \frac{9SH(H-h)\ln(3/\delta)}{h\epsilon}\right\} \leq \delta$.

These bounds show the approximate convergence resulted from the approximate model. However, the asymptotic performance gaps – both in terms of the regret and Uniform PAC – of $h$-RTDP-AM approach those experienced by an optimal policy of the approximate model. Interestingly, although $h$-RTDP-AM updates using the approximate model, while interacting with the true MDP, its convergence rate (to the asymptotic performance) is similar to that of $h$-RTDP (Theorem 4).

### 6.2 $h$-**RTDP with Approximate Value Updates ($h$-RTDP-AV)**

Another important question in the analysis of approximate DP algorithms is their performance under approximate value updates, motivated by the need to use function approximation. This is often modeled by an extra noise $|\epsilon_V(s)| \leq \epsilon_V$ added to the update rule [4]. Following this approach, we

study such perturbation in $h$-RTDP. Specifically, in $h$-RTDP-AV the value update rule is modified such that it contains an error term (see Algorithm 2),

$$\bar{V}_t^k(s_t^k) = \epsilon_V(s_t^k) + T^h \bar{V}_{h_c}^{k-1}(s_t^k).$$

For $\epsilon_V(s_t^k) = 0$, the exact $h$ is recovered. The pseudocode of $h$-RTDP-AV is supplied in Appendix 12.

Similar to the previous section, we follow the same proof technique as for Theorem 4 to establish the following performance bound for $h$-RTDP-AV (proof in Appendix 12).

**Theorem 6** (Performance of $h$-RTDP-AV). *Let $\epsilon, \delta > 0$. The following holds for $h$-RTDP-AV:*

1. *With probability $1 - \delta$, for all $K > 0$,* $\quad \mathrm{Regret}(K) \leq \frac{9SH(H-h)}{h}(1 + \frac{H}{h}\epsilon_V)\ln(\frac{3}{\delta}) + \frac{2H}{h}\epsilon_V K.$

2. *Let $\Delta_V = 2H\epsilon_V$. Then,* $\quad \mathrm{Pr}\left\{\exists \epsilon > 0 \; : \; N_\epsilon^{\frac{\Delta_V}{h}} \geq \frac{9SH(H-h)(1+\frac{\Delta_V}{2h})\ln(\frac{3}{\delta})}{h\epsilon}\right\} \leq \delta.$

As in Section 6.1, the results of Theorem 6 exhibit an asymptotic linear regret $O(H\epsilon_V K/h)$. As proven in Proposition 20 in Appendix 15, such performance gap exists in ADP with approximate value updates. Furthermore, the convergence rate in $S$ to the asymptotic performance of $h$-RTDP-AV is similar to that of its exact version (Theorem 4). Unlike in $h$-RTDP-AM, the asymptotic performance of $h$-RTDP-AV *improves* with $h$. This quantifies a clear benefit of using lookahead policies in online planning when the value function is approximate.

### 6.3 $h$-**RTDP with Approximate State Abstraction** ($h$-**RTDP-AA**)

We conclude the analysis of approximate $h$-RTDP with exploring the advantages of combining it with approximate state abstraction [1]. The central result of this section establishes that given an approximate state abstraction, $h$-RTDP converges with sample, computation, and space complexity *independent* of the size of the state space $S$, as long as $S_h^{Tot}$ is smaller than $S$ (i.e., when performing $h$-lookahead is $S$ independent, Remark 2). This is in contrast to the computational complexity of ADP in this setting, which is still $O(HSA)$ (see Appendix 15.3 for further discussion). State abstraction has been widely investigated in approximate planning [12, 11, 16, 1], as a means to deal with large state space problems. Among existing approximate abstraction settings, we focus on the following one. For any $n \in \{0\} \cup [\frac{H}{h} - 1]$, we define $\phi_{nh+1} : \mathcal{S} \to \mathcal{S}_\phi$ to be a mapping from the state space $\mathcal{S}$ to reduced space $\mathcal{S}_\phi$, $S_\phi = |\mathcal{S}_\phi| \ll S$. We make the following assumption:

**Assumption 1** (Approximate Abstraction, [23], definition 3.3). *For any $s, s' \in \mathcal{S}$ and $n \in \{0\} \cup [\frac{H}{h} - 1]$ for which $\phi_{nh+1}(s) = \phi_{nh+1}(s')$, we have $|V_{nh+1}^*(s) - V_{nh+1}^*(s')| \leq \epsilon_A.$*

Let us denote by $\{\bar{V}_{\phi,nh+1}^k\}_{n=0}^{H/h}$ the values stored in memory by $h$-RTDP-AA at the $k$'th episode. Unlike previous sections, the value function per time step contains $S_\phi$ entries, $\bar{V}_{\phi,1+nh}^k \in \mathbb{R}^{S_\phi}$. Note that if $\epsilon_A = 0$, then optimal value function can be represented in the reduced state space $\mathcal{S}_\phi$. However, if $\epsilon_A$ is positive, exact representation of $V^*$ is not possible. Nevertheless, the asymptotic performance of $h$-RTDP-AA will be 'close', up to error of $\epsilon_A$, to the optimal policy.

Furthermore, the definition of the multi-step Bellman operator (2) and $h$-greedy policy (3) should be revised, and with some abuse of notation, defined as

$$a_t^k \in \arg\max_{\pi_0(s_t^k)} \max_{\pi_1,\ldots,\pi_{t_c-1}} \mathbb{E}\left[\sum_{t'=0}^{t_c-1} r_{t'} + \bar{V}_{\phi,h_c}^{k-1}(\phi_{h_c}(s_{t_c})) \mid s_0 = s_t^k\right], \tag{6}$$

$$T_\phi^h \bar{V}_{\phi,h_c}^{k-1}(s_t^k) := \max_{\pi_0,\ldots,\pi_{h-1}} \mathbb{E}\left[\sum_{t'=0}^{h-1} r_{t'} + \bar{V}_{\phi,t+h}^{k-1}(\phi_{t+h}(s_h)) \mid s_0 = s_t^k\right]. \tag{7}$$

Eq. (6) and (7) indicate that similar to (3), the $h$-lookahead policy uses the given model to plan for $h$ time steps ahead. Differently from (3), the value after $h$ time steps is the one defined in the *reduced state* space $\mathcal{S}_\phi$. Note that the definition of the $h$-greedy policy for $h$-RTDP-AA in (6) is equivalent to the one used in Algorithm 8, obtained by similar recursion as for the optimal Bellman operator (2).

$h$-RTDP-AA modifies both the value update and the calculation of the $h$-lookahead policy (the value update and action choice in algorithm 2). The $h$-lookahead policy is replaced by $h$-lookahead defined in (6). The value update is substituted by (7), i.e, $\bar{V}_{\phi,t}^k(\phi_t(s_t^k)) = T_\phi^h \bar{V}_{\phi,h_c}^{k-1}(s_t^k)$. The full pseudocode of $h$-RTDP-AA is supplied in Appendix 13. By similar technique, as in the proof of Theorem 4, we establish the following performance guarantees to $h$-RTDP-AA (proof in Appendix 13).

| Setting | $h$-RTDP Regret (This work) | ADP Regret [4] | UCT |
|---|---|---|---|
| Exact (5) | $\tilde{\mathcal{O}}\big(SH(H-h)/h\big)$ | 0 | $\Omega(\exp(\exp(H)))$ [9] |
| App. Model (6.1) | $\tilde{\mathcal{O}}\big(SH(H-h)/h+\Delta_P K\big)$ | $\Delta_P K$ | N.A |
| App. Value (6.2) | $\tilde{\mathcal{O}}\big(SH(H-h)g^{\epsilon}_{H/h}/h+\Delta_V K/h\big)$ | $\Delta_V K/h$ | N.A |
| App. Abstraction (6.3) | $\tilde{\mathcal{O}}\big(S_\phi H(H-h)/h + \Delta_A K/h\big)$ | $\Delta_A K/h$ | N.A |

Table 1: The lookhead horizon is $h$ and the horizon of the MDP is $H$. We denote $g^{\epsilon}_{H/h} = (1 + H\epsilon_V/h)$, $\Delta_P = H(H-1)\epsilon_P$, $\Delta_V = 2H\epsilon_V$, and $\Delta_A = H\epsilon_A$. The table summarizes the regret bounds of the $h$-RTDP settings studied in this work and compares them to those of their corresponding ADP approaches. The performance of ADP is based on standard analysis, supplied in Propositions 19, 20, 21 in Appendix 15.

**Theorem 7** (Performance of $h$-RTDP-AA)**.** *Let $\epsilon, \delta > 0$. The following holds for $h$-RTDP-AA:*

*1. With probability $1 - \delta$, for all $K > 0$,* $\quad \mathrm{Regret}(K) \leq \frac{9S_\phi H(H-h)}{h} \ln(3/\delta) + \frac{H\epsilon_A}{h} K.$

*2. Let $\Delta_A = H\epsilon_A$. Then,* $\quad \mathrm{Pr}\left\{\exists \epsilon > 0 \; : \; N_\epsilon^{\frac{\Delta_A}{h}} \geq \frac{9S_\phi H(H-h)\ln(3/\delta)}{h\epsilon}\right\} \leq \delta.$

Theorem 7 establishes $S$-independent performance bounds that depend on the size of the reduced state space $S_\phi$. The asymptotic regret and Uniform PAC guarantees are approximate, as the state abstraction is approximate. Furthermore, they are improving with the quality of approximation $\epsilon_A$, i.e., their asymptotic gap is $O(H\epsilon_A/h)$ relative to the optimal policy. Moreover, the asymptotic performance of $h$-RTDP-AA improves as $h$ is increased. Importantly, since the computation complexity of each episode of $h$-RTDP is independent of $S$ (Section 3), the computation required to reach the approximate solution in $h$-RTDP-AA is also $S$-independent. This is in contrast to the computational cost of DP that depends on $S$ and is $O(SHA)$ (see Appendix 15.3 for further discussion).

## 7 Discussion and Conclusions

**RTDP vs. DP.** The results of Sections 5 and 6 established finite-time convergence guarantees for the exact $h$-RTDP and its three approximations. In the approximate settings, as expected, the regret has a linear term of the form $\Delta K$, where $\Delta$ is linear in the approximation errors $\epsilon_P$, $\delta$, and $\epsilon_A$, and thus, the performance is continuous in these parameters, as we would desire. We refer to $\Delta K$ as the *asymptotic regret*, since it dominates the regret as $K \to \infty$.

A natural measure to evaluate the quality of $h$-RTDP in the approximate settings is comparing its regret to that of its corresponding approximate DP (ADP). Table 1 summarizes the regrets of the approximate $h$-RTDPs studied in this paper and their corresponding ADPs. ADP calculates approximate values $\{V^*_{nh+1}\}_{n=0}^{H/h}$ by backward induction. Based on these values, the same $h$-lookahead policy by which $h$-RTDP acts is evaluated. In the analysis of ADP, we use standard techniques developed for the discounted case in [4]. From Table 1, we reach the following conclusion: *the asymptotic performance (in terms of regret) of approximate $h$-RTDP is equivalent to that of a corresponding approximate DP algorithm.* Furthermore, it is important to note that the asymptotic error decreases with $h$ for the approximate value updates and approximate abstraction settings for both RTDP and DP algorithms. In these settings, the error is caused by approximation in the value function. By increasing the lookahead horizon $h$, the algorithm uses less such values and relies more on the model which is assumed to be correct. Thus, the algorithm becomes less affected by the value function approximation.

**Conclusions.** In this paper, we formulated $h$-RTDP, a generalization of RTDP that acts by a lookahead policy, instead of by a 1-step greedy policy, as in RTDP. We analyzed the finite-sample performance of $h$-RTDP in its exact form, as well as in three approximate settings. The results indicate that $h$-RTDP converges in a very strong sense. Its regret is constant w.r.t. to the number of episodes, unlike in, e.g., reinforcement learning where a lower bound of $\tilde{\mathcal{O}}(\sqrt{SAHT})$ exists [2, 18]. Furthermore, the analysis reveals that the sample complexity of $h$-RTDP improves by increasing the lookahead horizon $h$ (Remark 3). Moreover, the asymptotic performance of $h$-RTDP was shown to be equivalent to that of ADP (Table 1), which under no further assumption on the approximation error, is the best we can hope for.

We believe this work opens interesting research venues, such as studying alternatives to the solution of the $h$-greedy policy (see Section 9), studying a Receding-Horizon extension of RTDP, RTDP with

function approximation, and formulating a Thompson-Sampling version of RTDP, as the standard RTDP is an 'optimistic' algorithm. As the analysis developed in this work was shown to be quite generic, we hope that it can assist with answering some of these questions. On the experimental side, more needs to be understood, especially comparing RTDP with MCTS and studying how RTDP can be combined with deep neural networks as the value function approximator.

## 8 Broader Impact

Online planning algorithms, such as $A^*$ and RTDP, have been extensively studied and applied in AI for well over two decades. Our work quantifies the benefits of using lookahead-policies in this class of algorithms. Although lookahead-policies have also been widely used in online planning algorithms, their theoretical justification was lacking. Our study sheds light on the benefits of lookahead-policies. Moreover, the results we provide in this paper suggest improved ways for applying lookahead-policies in online planning with benefits when dealing with various types of approximations. This work opens up the room for practitioners to improve their algorithms and base lookahead policies on solid theoretical ground.

**Acknowledgements.** We thank the reviewers for their helpful comments and feedback. Y.E. and S.M. were partially supported by the ISF under contract 2199/20.

## Footnotes

[6]The results can also be applied to time-dependent MDPs, however, the notations will be more involved.

[7]In fact, $h$-RTDP does not need to store $V_1$ and $V_{H+1}$, they are only used in the analysis.

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
