[Supplementary Material]

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

# 9 Per-Episode Complexity of $h$-RTDP

In this section, we define and analyze the *Forward-Backward DP* by which an $h$-greedy policy can be calculated from a current state $s_t^k$ according to (3). Observe that the algorithm is based on a 'local' information, i.e., it does not need access to the entire state space, but to a portion of the state space in the 'vicinity' of the current state $s_t^k$. Furthermore, it does not assume prior knowledge on this vicinity.

## 9.1 Forward-Backward Dynamic Programming Approach

---

**Algorithm 3** $h$-Forward-Backward DP

---
**Input:** $s$, transition $p$, reward $r$, lookahead horizon $h$, value at the end of lookahead horizon $\bar{V}$
$\{S_{t'}(s)\}_{t'=1}^{h+1}$ = Forward-Pass($s,p, h$)
action = Backward-Pass($\{S_{t'}(s)\}_{t'=1}^{h+1}, r, p, h, \bar{V}$)
**return:** action

---

---

**Algorithm 4** Forward-Pass

---
**Input:** Starting state $s$, $p$, $h$
**Init:** $\mathcal{S}_1 = \{s\}$, $\forall t' \in [h]/\{1\}$, $\mathcal{S}_{t'}(s) = \{\}$
**for** $t' = 2, 3, \ldots, h+1$ **do**
  **for** $s_{t'-1} \in \mathcal{S}_{t'-1}(s)$ **do**
    # acquire possible next states from $s_{t'-1}$
    **for** $a \in \mathcal{A}$ **do**
      $\mathcal{S}_{t'}(s) = \mathcal{S}_{t'}(s) \cup \{s' : p(s' \mid s, a) > 0\}$
    **end for**
  **end for**
**end for**
**return:** $\{\mathcal{S}_{t'}(s)\}_{t'=1}^{h+1}$

---

**Algorithm 5** Backward-Pass

---
**Input:** $\{\mathcal{S}_{t'}(s)\}_{t'=1}^{h+1}, r, p, h, \bar{V}$
# initialize values by arbitrary value $C$
**Init:** $\forall t' \in [h-1]$, $\forall s \in \mathcal{S}_{t'}(s)$, $V_{t'}(s) = C$
# Assign the value at $t' = h$ to the current value, $V$.
**for** $s \in \mathcal{S}_{h+1}(s)$ **do**
  $V_{h+1}(s) = \bar{V}(s)$
**end for**
**for** $t' = h, h-1, \ldots, 2$ **do**
  **for** $s \in \mathcal{S}_{t'}(s)$ **do**
    $V_{t'}(s) = \max_a r(s, a) + p(\cdot \mid s, a) V_{t'+1}$
  **end for**
**end for**
**return:** $\arg\max_a r(s, a) + p(\cdot \mid s, a) V_2$

---

The Forward-Backword DP (Algorithm 3) approach is built on the following observation: would we known the accessible state space from $s$ in next $h$ time steps we could use Backward Induction (i.e., Value Iteration) on a finite-horizon MDP, with an horizon of $h$, and calculate the optimal policy from $s$. Unfortunately, as we do not assume such a prior knowledge, we have to calculate this set before applying the backward induction step. Thus, Forward-Backward DP first build this set (in the first, 'Forward' stage) and later applies standard backward induction (in the 'Backward' stage). In Proposition 8, we establish that calculating the set of accessible states can be done efficiently

Let us first analyze the *computational complexity* of Algorithm 3 using the following definitions. Let $\mathcal{S}_{t'}(s)$ be the set of reachable states from state $s$ in $t'$ times steps, formally,

$$\mathcal{S}_{t'}(s) = \{s' \mid \exists \pi : p^\pi(s_{t'} = s' \mid s_0 = s, \pi) > 0\},$$

where $p^\pi(s_{t'} = s' \mid s_0 = s, \pi) = \mathbb{E}[\mathbb{1}\{s_{t'} = s'\} \mid s_0 = s, \pi]$. The cardinality of this set is denoted by $|\mathcal{S}_{t'}(s)|$. let $\mathcal{N} := \max_s |\mathcal{S}_2(s)|$ be the maximal number of accessible states in 1-step (maximal 'nearest neighbors' from any state). Furthermore, let the total reachable states in $h$ time steps from state $s$ be $S_h^{Tot}(s) = \sum_{t'=1}^{h} |\mathcal{S}_{t'}(s)|$. When $S_h^{Tot}(s)$ is small, as we establish in this section, local search up to an horizon of $h$ can be done efficiently with the Forward-Backward DP, unlike the exhaustive search approach.

Based on the above definitions we analyze the computational complexity of Forward-Backward DP starting from the Forward-Pass stage.

**Proposition 8** (Computation Cost of Forward-Pass)**.** *The Forward-Pass stage of FB-DP can be implemented with the computation cost of $O(\mathcal{N}A S_h^{Tot}(s))$.*

*Proof.* Calculating the set $\{s' : p(s' \mid s, a) > 0\}$ cost is upper bounded by $O(\mathcal{N})$ as we need to enumerate at most all possible $O(\mathcal{N})$ next-states. We assume that $\mathcal{S}_{t'} = \mathcal{S}_{t'}(s) \cup \{s' : p(s' \mid s, a) > 0\}$ can be done by $O(\mathcal{N})$, e.g., when using a hash-table for saving $\mathcal{S}_{t'}$ in memory. As we need to repeat this operation $A$ times, the complexity for each $t' \in \{2, 3, .., h + 1\}$ is upper bounded by $O(\mathcal{N}A|S_{t'-1}(s)|)$. Summing over all $t'$ we get that the computational complexity of the Forward pass is upper bounded by

$$O\left(\mathcal{N}A \sum_{t'=2}^{h+1} |S_{t'-1}(s)|\right) = O\big(\mathcal{N}A|S_h^{Tot}(s)|\big),$$

where the second equality holds by definition of total number of accessible states in $h$ time steps. □

The computational complexity of the backward passage is the computational complexity of Backward Induction, which is the total number of states in which actions can be taken times the number of actions per state, i.e.,

$$O(A\mathcal{N}S_h^{Tot}(s)).,\tag{8}$$

where the origin of the factor $\mathcal{N}$ is due to the need to calculate the sum $\sum_{s'} p(s' \mid s, a)V(s')$ for each $(s, a)$ pair, and, by definition, this sum contain at most $\mathcal{N}$ elements.

Using Proposition 8 and (8) we get that for every $t \in [H]$, the computational complexity of calculating an $h$-lookahead policy from a state $s$ using the Forward-Backward DP is bounded by,

$$O((\mathcal{N}A + \mathcal{N}A)S_h^{Tot}(s)) = O(\mathcal{N}AS_h^{Tot}),$$

where the last relation holds by definition, $S_h^{Tot} = \max_s S_h^{Tot}(s)$.

Finally, the *space complexity* of Forward-Backward DP is the space required the save in memory the possible visited states in $h$ time steps (their identity in the Forward-Pass and their values in the Backward-Pass). By definition it is at most $O(hS_h)$.

## 10   Real Time Dynamic Programming with Lookahead

This section contains the full proofs of all the results of Section 5 in chronological order.

**Lemma 2.** *For all $s \in \mathcal{S}$, $n \in \{0\} \cup [\frac{H}{h}]$, and $k \in [K]$, the value function of h-RTDP is (i) Optimistic:*
$V^*_{nh+1}(s) \leq \bar{V}^k_{nh+1}(s)$, *and (ii) Non-Increasing:* $\bar{V}^k_{nh+1}(s) \leq \bar{V}^{k-1}_{nh+1}(s)$.

*Proof.* Both claims are proven using induction.

*(i)* Let $n \in \{0\} \cup [\frac{H}{h}]$. By the initialization, $\forall s, n,\ V^*_{nh+1}(s) \leq V^0_{nh+1}(s)$. Assume the claim holds for the first $(k-1)$ episodes. Let $s^k_t$ be the state of the algorithm at a time step $t$ of the $k$'th episode at which a value update takes place, i.e., $t = nh + 1$, for some $n \in \{0\} \cup [\frac{H}{h}]$. By the value update of Algorithm 2 and (2), we have

$$\bar{V}^k_t(s^k_t) = (T^h \bar{V}^{k-1}_{h_c})(s^k_t) = (T^h \bar{V}^{k-1}_{t+h})(s^k_t) \geq (T^h V^*_{t+h})(s^k_t) = V^*_t(s^k_t).$$

The inequality holds by the induction hypothesis and the monotonicity of $T^h$, a consequence of the monotonicity of $T$, the optimal Bellman operator [4]. The last equality holds by the fact that the recursion is satisfied by the optimal value function (2). Thus, the induction step is proven for the first claim.

*(ii)* Let $n \in \{0\} \cup [\frac{H}{h}]$ and $t = nh + 1$ be a time step in which a value update takes place. To prove the base case, we use the optimistic initialization. Let $s^1_t$ be the state of the algorithm in the $t$'th time step of the first episode. By the update rule, we have

$$\bar{V}^1_t(s^1_t) = (T^h \bar{V}^0_{t+h})(s^0_t) = \max_{a_0, \ldots, a_{h-1}} \mathbb{E}\left[ \sum_{t'=0}^{h-1} r(s_{t'}, a_{t'}) + \bar{V}^0_{t+h}(s_h) \mid s_0 = s^0_t \right]$$

$$\overset{(a)}{\leq} h + H - (t + h - 1) = H - (t - 1) \overset{(b)}{=} \bar{V}^0_t(s^1_t).$$

**(a)** holds since $r(s, a) \in [0, 1]$ and by the optimistic initialization.
**(b)** observe that $H - (t - 1)$ is the value of the optimistic initialization.

Assume that the claim holds for the first $(k-1)$ episodes. Let $s^k_t$ be the state of the algorithm at a time step $t$ of the $k$'th episode at which a value update takes place, i.e., $t = nh+1$, for some $n \in \{0\} \cup [\frac{H}{h}]$. By the value update rule of Algorithm 2, we have $\bar{V}^k_t(s^k_t) = (T^h \bar{V}^{k-1}_{h_c})(s^k_t) = (T^h \bar{V}^{k-1}_{t+h})(s^k_t)$. If $s^k_t$ was previously updated, let $\bar{k}$ be the last episode in which the update occurred, i.e., $\bar{V}^{\bar{k}}_t(s^k_t) = (T^h \bar{V}^{\bar{k}-1}_{t+h})(s^k_t) = \bar{V}^{k-1}_t(s^k_t)$. By the induction hypothesis, we have that $\forall s, t,\ \bar{V}^{\bar{k}-1}_t(s) \geq \bar{V}^{k-1}_t(s)$. Using the monotonicity of $T^h$, we may write

$$\bar{V}^k_t(s^k_t) = (T^h \bar{V}^{k-1}_{t+h})(s^k_t) \leq (T^h \bar{V}^{\bar{k}-1}_{t+h})(s^k_t) = \bar{V}^{k-1}_t(s^k_t).$$

Thus, $\bar{V}^k_t(s^k_t) \leq \bar{V}^{k-1}_t(s^k_t)$ and the induction step is proved. If $s^k_t$ was not previously updated, then $\bar{V}^{k-1}_t(s^k_t) = \bar{V}^0_t(s^k_t)$. In this case, the induction hypothesis implies that $\forall s', \bar{V}^{k-1}_{t+h}(s') \leq \bar{V}^0_{t+h}(s')$ and the result is proven similarly to the base case. $\qquad\square$

**Lemma 3** (Optimality Gap and Expected Decrease)**.** *The expected cumulative value update at the $k$'th episode of h-RTDP satisfies* $\bar{V}^k_1(s^k_1) - V^{\pi_k}_1(s^k_1) = \sum_{n=1}^{\frac{H}{h}-1} \sum_{s \in \mathcal{S}} \bar{V}^{k-1}_{nh+1}(s) - \mathbb{E}[\bar{V}^k_{nh+1}(s) \mid \mathcal{F}_{k-1}].$

*Proof.* Let $n \in \{0\} \cup [\frac{H}{h}]$ and $t = nh + 1$ be a time step in which a value update takes place. By the definition of the update rule, the following holds for the value update at the visited state $s^k_t$:

$$\bar{V}^k_t(s^k_t) = (T^h \bar{V}^{k-1}_{t+h})(s^k_t) \tag{9}$$

$$= (T^{\pi_k(t)} \cdots T^{\pi_k(t+h-1)} \bar{V}^{k-1}_{t+h})(s^k_t) = \mathbb{E}\left[ \sum_{t'=t}^{t+h-1} r(s_{t'}, a_{t'}) + \bar{V}^{k-1}_{t+h}(s_{t+h}) \mid \pi_k, s_t = s^k_t \right]$$

$$\overset{(a)}{=} \mathbb{E}\left[ \sum_{t'=t}^{t+h-1} r(s^k_{t'}, a^k_{t'}) + \bar{V}^{k-1}_{t+h}(s^k_{t+h}) \mid \mathcal{F}_{k-1}, s^k_t \right]. \tag{10}$$

**(a)** We prove this passage for each reward element $r(s_{t'}, a_{t'})$ in the expectation. The proof for the expectation of $\bar{V}_{t+h}^{k-1}(s_{t+h})$ follows in a similar manner. Since the first expectation is w.r.t. the dynamics of the true model, a consequence of updating by the true model, for any $t' \geq t$, we may write

$$
\mathbb{E}\left[r(s_{t'}, a_{t'}) \mid \pi_k, s_t = s_t^k\right] = \sum_{s_{t'} \in \mathcal{S}} p(s_{t'} \mid s_t = s_t^k, \pi_k) r(s_{t'}, \pi_k(s_{t'}, t'))
$$

$$
\overset{(i)}{=} \sum_{s_{t'}^k \in \mathcal{S}} p(s_{t'}^k \mid s_t^k, \mathcal{F}_{k-1}) r(s_{t'}^k, \pi_k(s_{t'}^k, t')) = \mathbb{E}\left[r(s_{t'}^k, a_{t'}^k) \mid \mathcal{F}_{k-1}, s_t^k\right],
$$

where $p(s_{t'} \mid s_t^k, \pi_k)$ is the probability of starting at state $s_t^k$, following $\pi_k$, and reaching state $s_{t'}$ in $t' - t$ steps.

**(i)** We use the fact that $p(s_{t'} \mid s_t^k, \pi_k) = p(s_{t'}^k \mid s_t^k, \mathcal{F}_{k-1})$, in words, given the policy $\pi_k$ (which is $\mathcal{F}_{k-1}$ measurable) and $s_t^k$ the probability for a state $s_{t'}^k$ with $t' \geq t$ is independent of the rest of the history.

Now that we proved **(a)**, we take the conditional expectation of (9) w.r.t. $\mathcal{F}_{k-1}$ and use the tower rule to obtain

$$
\mathbb{E}\left[\bar{V}_t^k(s_t^k) \mid \mathcal{F}_{k-1}\right] = \mathbb{E}\left[\sum_{t'=t}^{t+h-1} r(s_{t'}^k, a_{t'}^k) + \bar{V}_{t+h}^{k-1}(s_{t+h}^k) \mid \mathcal{F}_{k-1}\right]. \tag{11}
$$

Summing (11) for all $n \in \{0\} \cup [\frac{H}{h}]$, and using the linearity of expectation and the fact that $\bar{V}_{H+1}^k(s) = 0$ for all $s, k$, we have

$$
\sum_{n=0}^{\frac{H}{h}-1} \mathbb{E}\left[\bar{V}_{nh+1}^k(s_{nh+1}^k) \mid \mathcal{F}_{k-1}\right] = \mathbb{E}\left[\sum_{t=1}^{H} r(s_t^k, a_t^k) \mid \mathcal{F}_{k-1}\right] + \sum_{n=1}^{\frac{H}{h}-1} \mathbb{E}\left[\bar{V}_{nh+1}^{k-1}(s_{nh+1}^k) \mid \mathcal{F}_{k-1}\right]
$$

$$
\iff \bar{V}_1^k(s_1^k) + \sum_{n=1}^{\frac{H}{h}-1} \mathbb{E}\left[\bar{V}_{nh+1}^k(s_t^k) \mid \mathcal{F}_{k-1}\right] = \mathbb{E}\left[\sum_{t=1}^{H} r(s_t^k, a_t^k) \mid \mathcal{F}_{k-1}\right] + \sum_{n=1}^{\frac{H}{h}-1} \mathbb{E}\left[\bar{V}_{nh+1}^{k-1}(s_{nh+1}^k) \mid \mathcal{F}_{k-1}\right]
$$

$$
\iff \bar{V}_1^k(s_1^k) + \sum_{n=1}^{\frac{H}{h}-1} \mathbb{E}\left[\bar{V}_{nh+1}^k(s_t^k) \mid \mathcal{F}_{k-1}\right] = V^{\pi_k}(s_1^k) + \sum_{n=1}^{\frac{H}{h}-1} \mathbb{E}\left[\bar{V}_{nh+1}^{k-1}(s_{nh+1}^k) \mid \mathcal{F}_{k-1}\right]
$$

$$
\iff \bar{V}_1^k(s_1^k) - V^{\pi_k}(s_1^k) = \sum_{n=1}^{\frac{H}{h}-1} \mathbb{E}\left[\bar{V}_{nh+1}^{k-1}(s_{nh+1}^k) - \bar{V}_{nh+1}^k(s_{nh+1}^k) \mid \mathcal{F}_{k-1}\right]. \tag{12}
$$

The second line holds by the fact that $s_1^k$ is measurable w.r.t. $\mathcal{F}_{k-1}$ The third line holds since

$$
V_1^{\pi_k}(s_1^k) = \mathbb{E}\left[\sum_{t=1}^{H} r(s_t^k, a_t^k) \mid s_1^k, \pi_k\right] = \mathbb{E}\left[\sum_{t=1}^{H} r(s_t^k, a_t^k) \mid \mathcal{F}_{k-1}\right].
$$

Applying Lemma 15 from Appendix 14 with $g_t^k = \bar{V}_t^k$ for $t = nh + 1$, we obtain

$$
(12) = \sum_{n=1}^{\frac{H}{h}-1} \sum_{s \in \mathcal{S}} \bar{V}_{nh+1}^{k-1}(s) - \mathbb{E}[\bar{V}_{nh+1}^k(s) \mid \mathcal{F}_{k-1}],
$$

which concludes the proof. Note that the update of $\bar{V}_t^k$ occurs only at the visited state $s_t^k$ and the update rule uses $\bar{V}_{t+h}^{k-1}$, i.e., it is measurable w.r.t. $\mathcal{F}_{k-1}$, and thus, it is valid to apply Lemma 15. □

**Theorem 4** (Performance of $h$-RTDP)**.** *Let $\epsilon, \delta > 0$. The following holds for $h$-RTDP:*

1. *With probability $1 - \delta$, for all $K > 0$,* $\quad \text{Regret}(K) \leq \frac{9SH(H-h)}{h} \ln(3/\delta)$.

2. $\Pr\left\{\exists \epsilon > 0 : N_\epsilon \geq \frac{9SH(H-h)\ln(3/\delta)}{h\epsilon}\right\} \leq \delta$.

*Proof.* We start by proving **Claim (1)**. We know that the following bounds hold on the regret:

$$\text{Regret}(K) := \sum_{k=1}^{K} V_1^*(s_1^k) - V_1^{\pi_k}(s_1^k) \overset{(a)}{\leq} \sum_{k=1}^{K} \bar{V}_1^k(s_1^k) - V_1^{\pi_k}(s_1^k)$$

$$\overset{(b)}{=} \sum_{k=1}^{K} \sum_{n=1}^{\frac{H}{h}-1} \sum_{s \in \mathcal{S}} \bar{V}_{nh+1}^{k-1}(s) - \mathbb{E}[\bar{V}_{nh+1}^k(s) \mid \mathcal{F}_{k-1}]. \tag{13}$$

**(a)** is by the optimism of the value function (Lemma 2), and **(b)** is by Lemma 3.

We would like to show that (13) is the regret of a Decreasing Bounded Process (DBP). We start by defining

$$X_k := \sum_{n=1}^{\frac{H}{h}-1} \sum_{s \in \mathcal{S}} \bar{V}_{nh+1}^k(s). \tag{14}$$

We now prove that $\{X_k\}_{k \geq 0}$ is a DBP. Note that $\{X_k\}_{k \geq 0}$

1. is decreasing, since $\forall s, t, \ \bar{V}_t^k(s) \leq \bar{V}_t^{k-1}(s)$ by Lemma 2, and thus, their sum is also decreasing, and

2. is bounded since $\forall s, t \ \bar{V}_t^k(s) \geq V_t^*(s) \geq 0$ by Lemma 2, and thus, the sum is bounded from below by 0.

We can show that the initial value $X_0$ is also bounded as

$$X_0 = \sum_{n=1}^{\frac{H}{h}-1} \sum_{s \in \mathcal{S}} \bar{V}_{nh+1}^0(s) \leq \sum_{n=1}^{\frac{H}{h}-1} \sum_{s \in \mathcal{S}} H = \frac{SH(H-h)}{h}.$$

Using the linearity of expectation and the definition (14), we observe that (13) can be written as

$$\text{Regret}(K) \leq (13) = \sum_{k=1}^{K} X_{k-1} - \mathbb{E}[X_k \mid \mathcal{F}_{k-1}],$$

which is regret of a DBP. Applying the bound on the regret of a DBP, Theorem 1, we conclude the proof of the first claim.

We now prove **Claim (2)**. Here we use a different technique than the one used in [15]. The technique allows us to prove uniform-PAC bounds for the approximate versions of $h$-RTDP described in Section 6. For these approximate versions, the uniform-PAC result is not a corollary of the regret bound and more careful analysis should be used.

For all $\epsilon > 0$, the following relations hold:

$$\mathbb{1}\{V_1^*(s_1^k) - V_1^{\pi_k}(s_1^k) \geq \epsilon\}\epsilon \overset{(a)}{\leq} \mathbb{1}\{\bar{V}_1^k(s_1^k) - V_1^{\pi_k}(s_1^k) \geq \epsilon\}\epsilon$$

$$\overset{(b)}{\leq} \mathbb{1}\{\bar{V}_1^k(s_1^k) - V_1^{\pi_k}(s_1^k) \geq \epsilon\}\left(\bar{V}_1^k(s_1^k) - V_1^{\pi_k}(s_1^k)\right)$$

$$\overset{(c)}{=} \mathbb{1}\{\bar{V}_1^k(s_1^k) - V_1^{\pi_k}(s_1^k) \geq \epsilon\}\left(\sum_{n=1}^{\frac{H}{h}-1} \sum_{s \in \mathcal{S}} \bar{V}_{nh+1}^{k-1}(s) - \mathbb{E}[\bar{V}_{nh+1}^k(s) \mid \mathcal{F}_{k-1}]\right)$$

$$\overset{(d)}{=} \mathbb{1}\{\bar{V}_1^k(s_1^k) - V_1^{\pi_k}(s_1^k) \geq \epsilon\}(X_{k-1} - \mathbb{E}[X_k \mid \mathcal{F}_{k-1}]). \tag{15}$$

**(a)** holds since for all $t, s$, $\bar{V}_t^k(s) \geq V_t^*(s)$ by Lemma 2. **(b)** holds by the indicator function. **(c)** holds by Lemma 3. **(d)** holds by the definition of $X_k$ from (14) and the linearity of expectation.

Let define $N_\epsilon(K) = \sum_{k=1}^{K} \mathbb{1}\{V_1^*(s_1^k) - V_1^{\pi_k}(s_1^k) \geq \epsilon\}$ as the number of times $V_1^*(s_1^k) - V_1^{\pi_k}(s_1^k) \geq \epsilon$ at the first $K$ episodes. For all $\epsilon > 0$, we may write

$$N_\epsilon(K)\epsilon \overset{(a)}{=} \sum_{k=1}^{K} \mathbb{1}\{V_1^*(s_1^k) - V_1^{\pi_k}(s_1^k) \geq \epsilon\}\epsilon \overset{(b)}{\leq} \sum_{k=1}^{K} \mathbb{1}\{\bar{V}_1^k(s_1^k) - V_1^{\pi_k}(s_1^k) \geq \epsilon\}(X_{k-1} - \mathbb{E}[X_k \mid \mathcal{F}_{k-1}])$$

$$\overset{(c)}{\leq} \sum_{k=1}^{K} X_{k-1} - \mathbb{E}[X_k \mid \mathcal{F}_{k-1}],$$

**(a)** holds by the definition of $N_\epsilon(K)$. **(b)** follows from (15). **(c)** holds because $\{X_k\}_{k\geq 0}$ is a DBP, and thus, $X_{k-1} - \mathbb{E}[X_k \mid \mathcal{F}_{k-1}] \geq 0$ a.s. Therefore, the following relation holds:

$$\left\{\forall K > 0 : \sum_{k=1}^{K} X_{k-1} - \mathbb{E}[X_k \mid \mathcal{F}_{k-1}] \leq \frac{9SH(H-h)}{h}\ln\frac{3}{\delta}\right\} \subseteq \left\{\forall \epsilon > 0 : N_\epsilon(K)\epsilon \leq \frac{9SH(H-h)}{h}\ln\frac{3}{\delta}\right\},$$

from which we obtain that for any $K > 0$,

$$\Pr\left(\forall \epsilon > 0 : N_\epsilon(K)\epsilon \leq \frac{9SH(H-h)}{h}\ln\frac{3}{\delta}\right) \geq \Pr\left(\forall K > 0 : \sum_{k=1}^{K} X_{k-1} - \mathbb{E}[X_k \mid \mathcal{F}_{k-1}] \leq \frac{9SH(Hh)}{h}\ln\frac{3}{\delta}\right) \overset{(a)}{\geq} 1 - \delta.$$

**(a)** holds because of the bound on the regret of DBP (see Theorem 1). Equivalently, for any $K > 0$,

$$\Pr\left(\exists \epsilon > 0 : N_\epsilon(K)\epsilon \geq \frac{9SH(H-h)}{h}\ln\frac{3}{\delta}\right) \leq \delta. \tag{16}$$

Note that for all $\epsilon > 0$, $K_1 \geq K_2$, $\mathbb{1}\{N_\epsilon(K_2)\epsilon \geq C\} = 1$ implies $\mathbb{1}\{N_\epsilon(K_1)\epsilon \geq C\} = 1$, and thus, $\mathbb{1}\{N_\epsilon(K)\epsilon \geq C\} \leq \lim_{K\to\infty} \mathbb{1}\{N_\epsilon(K)\epsilon \geq C\}$. Furthermore, $\mathbb{1}\{N_\epsilon(K)\epsilon \geq C\} \geq 0$ by definition. Thus, we can apply the Monotone Convergence Theorem to conclude the proof:

$$\Pr\left(\exists \epsilon > 0 : N_\epsilon\epsilon \geq \frac{9SH(H-h)}{h}\ln\frac{3}{\delta}\right) = \Pr\left(\lim_{K\to\infty}\left\{\exists \epsilon > 0 : N_\epsilon(K)\epsilon \geq \frac{9SH(H-h)}{h}\ln\frac{3}{\delta}\right\}\right)$$

$$= \mathbb{E}\left[\lim_{K\to\infty} \mathbb{1}\left\{\exists \epsilon > 0 : N_\epsilon(K)\epsilon \geq \frac{9SH(H-h)}{h}\ln\frac{3}{\delta}\right\}\right] \overset{(a)}{=} \lim_{K\to\infty} \mathbb{E}\left[\mathbb{1}\left\{\exists \epsilon > 0 : N_\epsilon(K)\epsilon \geq \frac{9SH(H-h)}{h}\ln\frac{3}{\delta}\right\}\right]$$

$$= \lim_{K\to\infty} \Pr\left(\exists \epsilon > 0 : N_\epsilon(K)\epsilon \geq \frac{9SH(H-h)}{h}\ln\frac{3}{\delta}\right) \overset{(b)}{\leq} \delta.$$

**(a)** is by the Monotone Convergence Theorem by which $\mathbb{E}[\lim_{k\to\infty} X_k] = \lim_{k\to\infty} \mathbb{E}[X_k]$, for $X_k \geq 0$ and $X_k \leq \lim_{k\to\infty} X_k$. **(b)** holds by (16). $\qquad\square$

# 11  $h$-RTDP with Approximate Model

---

**Algorithm 6** $h$-RTDP with Approximate Model ($h$-RTDP-AM)

---

    init: $\forall s \in \mathcal{S}$, $\forall n \in \{0\} \cup [\frac{H}{h}]$, $\bar{V}^0_{nh+1}(s) = H - nh$
    **for** $k \in [K]$ **do**
        Initialize $s^k_1$
        **for** $t \in [H]$ **do**
            **if** $(t-1) \mod h == 0$ **then**
                $h_c = t + h$
                $\bar{V}^k_t(s^k_t) = \hat{T}^h \bar{V}^{k-1}_{h_c}(s^k_t)$
            **end if**
            $a^k_t \in \arg\max_a r(s^k_t, a) + \hat{p}(\cdot|s^k_t, a)\hat{T}^{h_c - t - 1}\bar{V}^{k-1}_{h_c}$
            Act with $a^k_t$ and observe $s^k_{t+1} \sim p(\cdot \mid s^k_t, a^k_t)$
        **end for**
    **end for**

---

Algorithm 6 contains the pseudocode of $h$-RTDP with approximate model. The algorithm is exactly the same as $h$-RTDP (Algorithm 2) with the model $p$ and optimal Bellman operator $T$ replaced by their approximations $\hat{p}$ and $\hat{T}$. Meaning, $h$-RTDP is agnostic whether it uses the true or approximate model.

We now provide the full proofs of all results in Section 6.1 in their chronological order. We use the notation $\mathbb{E}_{\hat{P}}$ to denote expectation w.r.t. the approximate model, i.e., w.r.t. the dynamics $\hat{p}(s' \mid s, a)$ instead according to $p(s' \mid s, a)$.

**Lemma 9.** *For all $s \in \mathcal{S}$, $n \in \{0\} \cup [\frac{H}{h}]$, and $k \in [K]$:*

    *(i) Bounded / Optimism:* $\hat{V}^*_{nh+1}(s) \leq \bar{V}^k_{nh+1}(s)$.

    *(ii) Non-Increasing:* $\bar{V}^k_{nh+1}(s) \leq \bar{V}^{k-1}_{nh+1}(s)$.

*Proof.* Both claims are proven using induction.

*(i)* Let $n \in [0, \frac{H}{h} - 1]$ and denote $\hat{T}, \hat{V}^*$ as the optimal Bellman operators and optimal value of the approximate MDP $(\mathcal{S}, \mathcal{A}, \hat{p}, r, H)$. See that they satisfy usual Bellman equation 2.

By the initialization, $\forall s, t$, $\hat{V}^*_{1+hn}(s) \leq V^0_{1+hn}(s)$. Assume the claim holds for $k - 1$ episodes. Let $s^k_t$ be the state the algorithm is at in the $t = 1 + hn$ time step of the $k$'th episode, i.e., at a time step in which a value update is taking place. By the value update of Algorithm 6,

$$\bar{V}^k_t(s^k_t) = (\hat{T}^h \bar{V}_{t+h})(s^k_t) \geq (\hat{T}^h \hat{V}^*_{t+h})(s^k_t) = \hat{V}^*_t(s^k_t).$$

The second relation holds by the induction hypothesis and the monotonicity of $\hat{T}^h$, a consequence of the monotonicity of $\hat{T}$, the optimal Bellman operator [4]. The third relation holds by the recursion satisfied by the optimal value function (2). Thus, the induction step is proven for the first claim.

*(ii)* Let $n \in [0, \frac{H}{h} - 1]$ and let $t = 1 + hn$ be a time step in which a value update is taking place. To prove the base case of the second claim we use the optimistic initialization. Let $s^1_t$ be the state the algorithm is at in the $t$'th time step of the first episode. By the update rule,

$$\bar{V}^1_t(s^1_t) = (\hat{T}^h \bar{V}^0_{t+h})(s^0_t)$$

$$\overset{(1)}{=} \max_{\pi_0, \pi_1, .., \pi_{h-1}} \mathbb{E}_{\hat{P}'}\Big[\sum_{t'=0}^{h-1} r(s'_t, \pi_{t'}(s'_t)) + \bar{V}^0_{t+h}(s_h) \mid s_0 = s^0_t\Big]$$

$$\overset{(2)}{\leq} h + H - (t + h - 1) = H - (t - 1) \overset{(3)}{=} \bar{V}^0_t(s^1_t).$$

Relation $(1)$ is by the update rule (see Algorithm 6), when the expectation is taken place w.r.t. the approximate model $\hat{P}$. Relation $(2)$ holds since $r(s, a) \in [0, 1]$ and and by the optimistic initialization

(see that for $t$ the values at times step $t + h$ were not updated and keep their initial value). For (3) observe that $H - (t - 1)$ is the value of the optimistic initialization.

Assume the second claim holds for $k - 1$ episodes. Let $s_t^k$ be the state that the algorithm is at in the $t$'th time step of the $k$'th episode. Again, assume that $t = 1 + hn$, a time step in which a value update is being done. By the value update of Algorithm 6, we have

$$\bar{V}_t^k(s_t^k) = (\hat{T}^h \bar{V}_{t+h}^{k-1})(s_t^k).$$

If $s_t^k$ was previously updated, let $\bar{k}$ be the previous episode in which the update occured. By the induction hypothesis, we have that $\forall s, t, \ \bar{V}_t^{\bar{k}}(s) \geq \bar{V}_t^{k-1}(s)$. Using the monotonicity of $T^h$ (due to the monotonicity of the Bellman operator),

$$(\hat{T}^h \bar{V}_{t+h}^{k-1})(s_t^k) \leq (\hat{T}^h \bar{V}_{t+h}^{\bar{k}})(s_t^k) = \bar{V}_t^{k-1}(s_t^k).$$

Thus, $\bar{V}_t^k(s_t^k) \leq \bar{V}^{k-1}(s_t^k)$ and the induction step is proved. If $s_t^k$ was not previously updated, then $\bar{V}_t^{k-1}(s_t^k) = \bar{V}_t^0(s_t^k)$. In this case, the induction hypothesis implies that $\forall s', \bar{V}_{t+h}^{k-1}(s') \leq \bar{V}_{t+h}^0(s')$ and the result is proven similarly to the base case. $\qquad\square$

**Lemma 10.** *The expected cumulative value update at the $k$'th episode of h-RTDP-AM satisfies the following relation:*

$$\bar{V}_1^k(s_1^k) - V_1^{\pi_k}(s_1^k) = \frac{H(H-1)}{2}\epsilon_P$$

$$+ \sum_{n=1}^{\frac{H}{h}-1} \sum_{s \in \mathcal{S}} \bar{V}_{nh+1}^{k-1}(s) - \mathbb{E}[\bar{V}_{nh+1}^k(s) \mid \mathcal{F}_{k-1}].$$

*Proof.* Let $n \in [0, \frac{H}{h} - 1]$ and let $t = 1 + hn$ be a time step in which a value update is taking place. By the definition of the update rule, the following holds for the update at the visited state $s_t^k$:

$$\bar{V}_t^k(s_t^k) = (\hat{T}^h \bar{V}_{t+h}^{k-1})(s_t^k) \tag{17}$$

$$= (\hat{T}^{\pi_k(t)} \cdots \hat{T}^{\pi_k(t+h-1)} \bar{V}_{t+h}^{k-1})(s_t^k)$$

$$= \mathbb{E}_{P'}\left[ \sum_{t'=t}^{t+h-1} r(s_{t'}^k, a_{t'}^k) + \bar{V}_{t+h}^{k-1}(s_{t+h}^k) \mid \pi_k, s_t^k \right]$$

$$= \mathbb{E}\left[ \sum_{t'=t}^{t+h-1} r(s_{t'}^k, a_{t'}^k) + \bar{V}_{t+h}^{k-1}(s_{t+h}^k) \mid \pi_k, s_t^k \right]$$

$$+ \mathbb{E}_{P'}\left[ \sum_{t'=t}^{t+h-1} r(s_{t'}^k, a_{t'}^k) + \bar{V}_{t+h}^{k-1}(s_{t+h}^k) \mid \pi_k, s_t^k \right] - \mathbb{E}\left[ \sum_{t'=t}^{t+h-1} r(s_{t'}^k, a_{t'}^k) + \bar{V}_{t+h}^{k-1}(s_{t+h}^k) \mid \pi_k, s_t^k \right]$$

$$= \mathbb{E}\left[ \sum_{t'=t}^{t+h-1} r(s_{t'}^k, a_{t'}^k) + \bar{V}_{t+h}^{k-1}(s_{t+h}^k) \mid \pi_k, s_t^k \right]$$

$$+ \sum_{t'=t}^{t+h-1} \sum_{s_{t'}} \left( P^{\pi_k}(s_{t'} \mid s_t^k) - \hat{P}^{\pi_k}(s_{t'} \mid s_t^k) \right) r(s_{t'}^k, a_{t'}^k) + \sum_{s_{t+h}} \left( P^{\pi_k}(s_{t+h} \mid s_t^k) - \hat{P}^{\pi_k}(s_{t+h} \mid s_t^k) \right) \bar{V}_{t+h}^{k-1}(s_{t+h}^k))$$

$$\leq \mathbb{E}\left[ \sum_{t'=t}^{t+h-1} r(s_{t'}^k, a_{t'}^k) + \bar{V}_{t+h}^{k-1}(s_{t+h}^k) \mid \pi_k, s_t^k \right]$$

$$+ \sum_{t'=t}^{t+h-1} \sum_{s_{t'}} \left| P^{\pi_k}(s_{t'} \mid s_t^k) - \hat{P}^{\pi_k}(s_{t'} \mid s_t^k) \right| + (H - (t+h-1)) \sum_{s_{t+h}} \left| P^{\pi_k}(s_{t+h} \mid s_t^k) - \hat{P}^{\pi_k}(s_{t+h} \mid s_t^k) \right|.$$

Applying Lemma 16 we bound the above by,

$$(17) \leq \mathbb{E}\left[\sum_{t'=t}^{t+h-1} r(s_{t'}^k, a_{t'}^k) + \bar{V}_{t+h}^{k-1}(s_{t+h}^k) \mid \pi_k, s_t^k\right] + \sum_{t'=t}^{t+h-1} (t'-t)\epsilon_P + (H - (t+h-1))h\epsilon_P$$

$$= \mathbb{E}\left[\sum_{t'=t}^{t+h-1} r(s_{t'}^k, a_{t'}^k) + \bar{V}_{t+h}^{k-1}(s_{t+h}^k) \mid \pi_k, s_t^k\right] - \frac{1}{2}(h-1)h\epsilon_P + (H-t)h\epsilon_P$$

$$= \mathbb{E}\left[\sum_{t'=t}^{t+h-1} r(s_{t'}^k, a_{t'}^k) + \bar{V}_{t+h}^{k-1}(s_{t+h}^k) \mid \mathcal{F}_{k-1}, s_t^k\right] - \frac{1}{2}(h-1)h\epsilon_P + (H-t)h\epsilon_P. \qquad (18)$$

Where the second relation holds by using the close form of the arithmetic sum and by algebraic manipulations. For the third relation, we observe that given $\pi_k, s_t^k$ the state $s_{t'}^k$ with $t' \geq t$ is independent of the past episodes (see 10),

$$\mathbb{E}\left[\sum_{t'=t}^{t+h-1} r(s_{t'}^k, a_{t'}^k) + \bar{V}_{t+h}^{k-1}(s_{t+h}^k) \mid \pi_k, s_t^k\right] = \mathbb{E}\left[\sum_{t'=t}^{t+h-1} r(s_{t'}^k, a_{t'}^k) + \bar{V}_{t+h}^{k-1}(s_{t+h}^k) \mid \mathcal{F}_{k-1}, s_t^k\right]$$

Taking the conditional expectation w.r.t. $\mathcal{F}_{k-1}$ of both (17) and its RHS (18), using the tower property and the fact for all $s$, $\bar{V}_{H+1}(s) = 0$ we get,

$$\mathbb{E}\left[\bar{V}_t^k(s_t^k) \mid \mathcal{F}_{k-1}\right] \leq \mathbb{E}\left[\sum_{t'=t}^{t+h-1} r(s_{t'}^k, a_{t'}^k) + \bar{V}_{t+h}^{k-1}(s_{t+h}^k) \mid \mathcal{F}_{k-1}\right]$$
$$- \frac{1}{2}(h-1)h\epsilon_P + (H-t)h\epsilon_P$$

Let us denote $d_n := -\frac{1}{2}(h-1)h\epsilon_P + (H-n)h\epsilon_P$. Summing the above relation for all $n \in [\frac{H}{h}] - 1$, using linearity of expectation, and the fact $\bar{V}_{H+1}^k(s) = $ for all $s, k$,

$$\sum_{n=0}^{\frac{H}{h}-1} \mathbb{E}\left[\bar{V}_{1+nh}^k(s_t^k) \mid \mathcal{F}_{k-1}\right] = \mathbb{E}\left[\sum_{t=1}^{H} r(s_t^k, a_t^k) \mid \mathcal{F}_{k-1}\right] + \sum_{n=1}^{\frac{H}{h}-1} \mathbb{E}\left[\bar{V}_{1+nh}^{k-1}(s_{1+nh}^k) \mid \mathcal{F}_{k-1}\right] + \sum_{n=0}^{\frac{H}{h}-1} d_{1+nh}$$
$$(19)$$

By simple algebraic manipulation we get $\sum_{n=0}^{\frac{H}{h}-1} d_{1+nh} = \frac{1}{2}H(H-1)\epsilon_P$ (see Lemma 18). Thus, (19) has the following equivalent forms, by which we conclude the proof of this lemma.

$$\iff \bar{V}_1^k(s_1^k) + \sum_{n=1}^{\frac{H}{h}-1} \mathbb{E}\left[\bar{V}_{1+nh}^k(s_t^k) \mid \mathcal{F}_{k-1}\right] = \mathbb{E}\left[\sum_{t=1}^{H} r(s_t^k, a_t^k) \mid \mathcal{F}_{k-1}\right] + \sum_{n=1}^{\frac{H}{h}-1} \mathbb{E}\left[\bar{V}_{1+nh}^{k-1}(s_{1+nh}^k) \mid \mathcal{F}_{k-1}\right] + \frac{1}{2}H(H-1)\epsilon_P$$

$$\iff \bar{V}_1^k(s_1^k) + \sum_{n=1}^{\frac{H}{h}-1} \mathbb{E}\left[\bar{V}_{1+nh}^k(s_t^k) \mid \mathcal{F}_{k-1}\right] = V^{\pi_k}(s_1^k) + \sum_{n=1}^{\frac{H}{h}-1} \mathbb{E}\left[\bar{V}_{1+nh}^{k-1}(s_{1+nh}^k) \mid \mathcal{F}_{k-1}\right] + \frac{1}{2}H(H-1)\epsilon_P$$

$$\iff \bar{V}_1^k(s_1^k) - V^{\pi_k}(s_1^k) = \sum_{n=1}^{\frac{H}{h}-1} \mathbb{E}\left[\bar{V}_{1+nh}^{k-1}(s_{1+nh}^k) - \bar{V}_{1+nh}^k(s_{1+nh}^k) \mid \mathcal{F}_{k-1}\right] + \frac{1}{2}H(H-1)\epsilon_P$$

$$\iff \bar{V}_1^k(s_1^k) - V^{\pi_k}(s_1^k) = \sum_{k=1}^{K}\sum_{n=1}^{\frac{H}{h}-1}\sum_{s} \bar{V}_{nh+1}^{k-1}(s) - \mathbb{E}[\bar{V}_{nh+1}^k(s) \mid \mathcal{F}_{k-1}] + \frac{1}{2}H(H-1)\epsilon_P$$

The second line holds by the fact $s_1^k$ is measurable w.r.t. $\mathcal{F}_{k-1}$, the third line holds since

$$V_1^{\pi_k}(s_1^k) = \mathbb{E}\left[\sum_{t=1}^{H} r(s_t^k, a_t^k) \mid \mathcal{F}_{k-1}\right].$$

The forth line holds by Lemma 15 with $\bar{V}_t^k = g_t^k$ for $t = nh + 1$. See that the update of $\bar{V}_t^k$ occurs only at the visited state $s_t^k$ and the update rule uses $\bar{V}_{t+1}^{k-1}$, i.e., it is measurable w.r.t. to $\mathcal{F}_{k-1}$, and it is valid to apply the lemma.

$\square$

**Theorem 5** (Performance of $h$-RTDP-AM). *Let $\epsilon, \delta > 0$. The following holds for $h$-RTDP-AM:*

*1. With probability $1 - \delta$, for all $K > 0$,* $\text{Regret}(K) \le \frac{9SH(H-h)}{h} \ln(3/\delta) + H(H-1)\epsilon_P K.$

*2. Let $\Delta_P = H(H-1)\epsilon_P$. Then,* $\Pr\left\{\exists \epsilon > 0 \; : \; N_\epsilon^{\Delta_P} \ge \frac{9SH(H-h)\ln(3/\delta)}{h\epsilon}\right\} \le \delta.$

*Proof.* We start by proving **claim (1)**. The following bounds on the regret hold.

$$
\begin{aligned}
\text{Regret}(K) &:= \sum_{k=1}^{K} V_1^*(s_1^k) - V_1^{\pi_k}(s_1^k) \\
&\le \sum_{k=1}^{K} \hat{V}_1^*(s_1^k) - V_1^{\pi_k}(s_1^k) + \frac{H(H-1)}{2}\epsilon_P \\
&\le \sum_{k=1}^{K} \bar{V}_1^k(s_1^k) - V_1^{\pi_k}(s_1^k) + \frac{H(H-1)}{2}\epsilon_P \\
&= H(H-1)\epsilon_P K + \sum_{k=1}^{K} \sum_{n=1}^{\frac{H}{h}-1} \sum_s \bar{V}_{nh+1}^{k-1}(s) - \mathbb{E}[\bar{V}_{nh+1}^k(s) \mid \mathcal{F}_{k-1}]
\end{aligned}
\tag{20}
$$

The second relation holds by Lemma 17 which relates the optimal value of the approximate model to the optimal value of the environment. The third relation is by the optimism of the value function (Lemma 9), and the forth relation is by Lemma 10.

We now observe the regret is a regret of a Decreasing Bounded Process. Let

$$
X_k := \sum_{n=1}^{\frac{H}{h}-1} \sum_s \bar{V}_{nh+1}^k(s),
\tag{21}
$$

and observe that $\{X_k\}_{g \ge 0}$ is a Decreasing Bounded Process.

1. It is decreasing since for all $s, t$ $\bar{V}_t^k(s) \le \bar{V}_t^{k-1}(s)$ by Lemma 9. Thus, their sum is also decreasing.

2. It is bounded since for all $s, t$ $\bar{V}_t^k(s) \ge V_t^*(s) \ge 0$ by Lemma 9. Thus, the sum is bounded from below by $0$.

See that the initial value can be bounded as follows,

$$
X_0 = \sum_{n=1}^{\frac{H}{h}-1} \sum_s \bar{V}_{nh+1}^0(s) \le \sum_{n=1}^{\frac{H}{h}-1} \sum_s H = \frac{SH(H-h)}{h}.
$$

Using linearity of expectation and the definition (21) we observe that (20) can be written,

$$
\text{Regret}(K) \le (20) = H(H-1)\epsilon_P K + \sum_{k=1}^{K} X_{k-1} - \mathbb{E}[X_k \mid \mathcal{F}_{k-1}],
$$

which is regret of A Bounded Decreasing Process. Applying the regret bound on DBP, Theorem 1, we conclude the proof of the first claim.

We now prove the **claim (2)** using the proving technique at Theorem 4. Denote $\Delta_P = H(H-1)\epsilon_P$. The following relations hold for all $\epsilon > 0$.

$$\mathbb{1}\left\{\hat{V}_1^*(s_1^k) - V_1^{\pi_k}(s_1^k) \geq \frac{\Delta_P}{2} + \epsilon\right\}\left(\epsilon + \frac{\Delta_P}{2}\right)$$

$$\leq \mathbb{1}\left\{\bar{V}_1^k(s_1^k) - V_1^{\pi_k}(s_1^k) \geq \frac{\Delta_P}{2} + \epsilon\right\}\left(\epsilon + \frac{\Delta_P}{2}\right)$$

$$\leq \mathbb{1}\left\{\bar{V}_1^k(s_1^k) - V_1^{\pi_k}(s_1^k) \geq \frac{\Delta_P}{2} + \epsilon\right\}\left(\bar{V}_1^k(s_1^k) - V_1^{\pi_k}(s_1^k)\right)$$

$$= \mathbb{1}\left\{\bar{V}_1^k(s_1^k) - V_1^{\pi_k}(s_1^k) \geq \frac{\Delta_P}{2} + \epsilon\right\}\left(\sum_{n=1}^{\frac{H}{h}-1}\sum_s \bar{V}_{nh+1}^{k-1}(s) - \mathbb{E}[\bar{V}_{nh+1}^k(s) \mid \mathcal{F}_{k-1}] + \frac{\Delta_P}{2}\right)$$

$$= \mathbb{1}\left\{\bar{V}_1^k(s_1^k) - V_1^{\pi_k}(s_1^k) \geq \frac{\Delta_P}{2} + \epsilon\right\}\left(X_{k-1} - \mathbb{E}[X_k \mid \mathcal{F}_{k-1}] + \frac{\Delta_P}{2}\right).$$

The first relation holds since for all $t, s$, $\bar{V}_t^k(s) \geq \hat{V}_t^*(s)$ by Lemma 9. The second relation holds by the indicator function and the third relation holds by Lemma 10. The forth relation holds by the definition of $X_k$ (21) and linearity of expectation. Using an algebraic manipulation the above leads to the following relation,

$$\mathbb{1}\left\{\hat{V}_1^*(s_1^k) - V_1^{\pi_k}(s_1^k) \geq \frac{\Delta_P}{2} + \epsilon\right\}\epsilon \leq \mathbb{1}\left\{\bar{V}_1^k(s_1^k) - V_1^{\pi_k}(s_1^k) \geq \frac{\Delta_P}{2} + \epsilon\right\}(X_{k-1} - \mathbb{E}[X_k \mid \mathcal{F}_{k-1}]) \tag{22}$$

As we wish the final performance to be compared to $V^*$ and not $\hat{V}$ we use the the first claim of Lemma 17, by which for all $s$, $\hat{V}_1^*(s) \geq V_1^*(s) - \frac{\Delta_P}{2}$. This implies that

$$\mathbb{1}\left\{V_1^*(s_1^k) - V_1^{\pi_k}(s_1^k) \geq \Delta_P + \epsilon\right\} \leq \mathbb{1}\left\{\hat{V}_1^*(s_1^k) - V_1^{\pi_k}(s_1^k) \geq \frac{\Delta_P}{2} + \epsilon\right\}. \tag{23}$$

Combining all the above, we get

$$\mathbb{1}\left\{V_1^*(s_1^k) - V_1^{\pi_k}(s_1^k) \geq \Delta_P + \epsilon\right\}\epsilon$$

$$\leq \mathbb{1}\left\{\hat{V}_1^*(s_1^k) - V_1^{\pi_k}(s_1^k) \geq \frac{\Delta_P}{2} + \epsilon\right\}\epsilon$$

$$\leq \mathbb{1}\left\{\bar{V}_1^k(s_1^k) - V_1^{\pi_k}(s_1^k) \geq \frac{\Delta_P}{2} + \epsilon\right\}(X_{k-1} - \mathbb{E}[X_k \mid \mathcal{F}_{k-1}]). \tag{24}$$

The first relation is by (23) and the second relation by (22).

Define $N_\epsilon(K) = \sum_{k=1}^K \mathbb{1}\{V_1^*(s_1^k) - V_1^{\pi_k}(s_1^k) \geq \Delta_P + \epsilon\}$ as the number of times $V_1^*(s_1^k) - V_1^{\pi_k}(s_1^k) \geq \Delta_P + \epsilon$ at the first $K$ episodes. Summing the above inequality (24) for all $k \in [K]$ and denote we get that for all $\epsilon > 0$

$$N_\epsilon(K)\epsilon = \sum_{k=1}^K \mathbb{1}\left\{V_1^*(s_1^k) - V_1^{\pi_k}(s_1^k) \geq \Delta_P + \epsilon\right\}\epsilon$$

$$\leq \sum_{k=1}^K \mathbb{1}\left\{\bar{V}_1^k(s_1^k) - V_1^{\pi_k}(s_1^k) \geq \frac{\Delta_P}{2} + \epsilon\right\}(X_{k-1} - \mathbb{E}[X_k \mid \mathcal{F}_{k-1}])$$

$$\leq \sum_{k=1}^K X_{k-1} - \mathbb{E}[X_k \mid \mathcal{F}_{k-1}].$$

The first relation holds by definition, the second by (24) and the third relation holds as $\{X_k\}_{k\geq0}$ is a DBP (21) and, thus, $X_{k-1} - \mathbb{E}[X_k \mid \mathcal{F}_{k-1}] \geq 0$ a.s. . Thus, the following relation holds

$$\left\{\forall K > 0 : \sum_{k=1}^K X_{k-1} - \mathbb{E}[X_k \mid \mathcal{F}_{k-1}] \leq \frac{9SH(H-h)}{h}\ln\frac{3}{\delta}\right\} \subseteq \left\{\forall \epsilon > 0 : N_\epsilon(K)\epsilon \leq \frac{9SH(H-h)}{h}\ln\frac{3}{\delta}\right\},$$

from which we get that for any $K > 0$

$$\Pr\left(\forall \epsilon > 0 : N_\epsilon(K)\epsilon \leq \frac{9SH(H-h)}{h} \ln \frac{3}{\delta}\right)$$

$$\geq \Pr\left(\forall K > 0 : \sum_{k=1}^{K} X_{k-1} - \mathbb{E}[X_k \mid \mathcal{F}_{k-1}] \leq \frac{9SH(Hh)}{h} \ln \frac{3}{\delta}\right) \geq 1 - \delta,$$

and the third relation holds the bound on the regret of DBP, Theorem 1. Equivalently, for any $K > 0$,

$$\Pr\left(\exists \epsilon > 0 : N_\epsilon(K)\epsilon \geq \frac{9SH(H-h)}{h} \ln \frac{3}{\delta}\right) \leq \delta. \tag{25}$$

Applying the Monotone Convergence Theorem as in the proof of Theorem 4 we conclude the proof.

$\square$

**Algorithm 7** $h$-RTDP with Approximate Value Updates ($h$-RTDP-AV)

---
init: $\forall s \in \mathcal{S}, \ n \in \{0\} \cup [\frac{H}{h}], \ \bar{V}^0_{nh+1}(s) = H - nh$
**for** $k \in [K]$ **do**
    Initialize $s^k_1$
    **for** $t \in [H]$ **do**
        **if** $(t-1) \mod h == 0$ **then**
            $h_c = t + h$
            $\bar{V}^k_t(s^k_t) = \epsilon_V(s^k_t) + T^h \bar{V}^{k-1}_{h_c}(s^k_t)$ ;        $\bar{V}^k_t(s^k_t) \leftarrow \min\{\bar{V}^k_t(s^k_t), \bar{V}^{k-1}_t(s^k_t)\}$ ;
        **end if**
        $a^k_t \in \arg\max_a r(s^k_t, a) + p(\cdot | s^k_t, a) T^{h_c - t - 1} \bar{V}^{k-1}_{h_c}$ ;
        Act with $a^k_t$ and observe $s^k_{t+1} \sim p(\cdot \mid s^k_t, a^k_t)$
    **end for**
**end for**

---

## 12   $h$-RTDP with Approximate Value updates

**Lemma 11.** *For all $s \in \mathcal{S}$, $n \in \{0\} \cup [\frac{H}{h}]$, and $k \in [K]$:*

    *(i) Bounded / Optimism:*

$$V^*_{nh+1}(s) \leq \bar{V}^k_{nh+1}(s) + \epsilon_V(\frac{H}{h} - n).$$

    *(ii) Non-Increasing:* $\bar{V}^k_{nh+1}(s) \leq \bar{V}^{k-1}_{nh+1}(s)$.

*Proof.* We prove the first claim by induction. The second claim holds by construction.

*(i)* Let $n \in \{0\} \cup [\frac{H}{h}]$. By the optimistic initialization, $\forall s, n, \ V^*_{1+hn}(s) - \epsilon_V(\frac{H}{h} - n) \leq V^*_{1+hn}(s) \leq V^0_{1+hn}(s)$. Assume the claim holds for $k-1$ episodes. Let $s^k_t$ be the state the algorithm is at in the $t = 1 + hn$ time step of the $k$'th episode, i.e., at a time step in which a value update is taking place. Let $e \in \mathbb{R}^S$ be the constant vector of ones. By the value update of Algorithm 7,

$$\bar{V}^k_t(s^k_t) = \min\{\epsilon_V(s^k_t) + T^h \bar{V}^{k-1}_{h_c}(s^k_t), \bar{V}^{k-1}_t(s^k_t)\}. \tag{26}$$

If the minimal value is $\bar{V}^{k-1}_t(s^k_t)$ then $\bar{V}^k_t(s^k_t)$ satisfies the induction hypothesis by the induction assumption. If $\epsilon_V(s^k_t) + T^h \bar{V}^{k-1}_{h_c}(s^k_t)$ is the minimal value in (26), then the following relation holds,

$$
\begin{aligned}
\bar{V}^k_t(s^k_t) &= \epsilon_V(s^k_t) + T^h \bar{V}^{k-1}_{t+h}(s^k_t) \\
&\geq -\epsilon_V + T^h \bar{V}^{k-1}_{t+h}(s^k_t) \\
&\geq -\epsilon_V + T^h \left( V^*_{t+h} - e\epsilon_V(\frac{H}{h} - n - 1) \right)(s^k_t) \\
&= -\epsilon_V + T^h V^*_{t+h}(s^k_t) - \epsilon_V(\frac{H}{h} - n - 1) \\
&= T^h V^*_{t+h}(s^k_t) - \epsilon_V(\frac{H}{h} - n) \\
&= V^*_t(s^k_t) - \epsilon_V(\frac{H}{h} - n).
\end{aligned}
$$

The second relation holds by the assumption $|\epsilon_V(s^k_t)| \leq \epsilon_V$. The third relation by the induction hypothesis and the monotonicity of $T^h$. The forth relation holds since for any constant $\alpha \in \mathbb{R}$ and $V \in \mathbb{R}^s$, $T(V + \alpha e) = TV + \alpha$ (e.g.,[4]) and thus $T^h(V + \alpha e) = T^h V + \alpha$. Lastly, the fifth relation holds by the Bellman equations (2).

*(ii)* The second claim holds by construction of the update rule $\bar{V}^k_t(s^k_t) \leftarrow \min\{\bar{V}^k_t(s^k_t), \bar{V}^{k-1}_t(s^k_t)\}$ which enforces $\bar{V}^k_t(s) \leq \bar{V}^{k-1}_t(s)$ for every updated state, and thus for all $s$ and $t$. $\qquad\square$

**Lemma 12.** *The expected cumulative value update at the $k$'th episode of $h$-RTDP-AV satisfies the following relation:*

$$\bar{V}_1^k(s_1^k) - V_1^{\pi_k}(s_1^k)$$

$$\leq \frac{H}{h}\epsilon_V + \sum_{k=1}^{K}\sum_{n=1}^{\frac{H}{h}-1}\sum_{s\in\mathcal{S}}\bar{V}_{nh+1}^{k-1}(s) - \mathbb{E}[\bar{V}_{nh+1}^k(s)\mid\mathcal{F}_{k-1}].$$

*Proof.* Let $n \in \{0\}\cup[\frac{H}{h}-1]$ and let $t = 1+hn$ be a time step in which a value update is taking place. By the definition of the update rule, the following holds for the update at the visited state $s_t^k$:

$$\bar{V}_t^k(s_t^k) = \epsilon_V(s_t^k) + (T^h\bar{V}_{t+h}^{k-1})(s_t^k)$$

$$\leq \epsilon_V + (T^{\pi_k(t)}\cdots T^{\pi_k(t+h-1)}\bar{V}_{t+h}^{k-1})(s_t^k)$$

$$= \epsilon_V + \mathbb{E}\left[\sum_{t'=t}^{t+h-1}r(s_{t'}^k,a_{t'}^k) + \bar{V}_{t+h}^{k-1}(s_{t+h}^k)\mid\mathcal{F}_{k-1},s_t^k\right].$$

Where the third relation holds by the same argument as in (10). Taking the conditional expectation w.r.t. $\mathcal{F}_{k-1}$, using the tower property and the fact for all $s$, $\bar{V}_{H+1}(s) = 0$ we get,

$$\mathbb{E}\left[\bar{V}_t^k(s_t^k)\mid\mathcal{F}_{k-1}\right] \leq \epsilon_V + \mathbb{E}\left[\sum_{t'=t}^{t+h-1}r(s_{t'}^k,a_{t'}^k) + \bar{V}_{t+h}^{k-1}(s_{t+h}^k)\mid\mathcal{F}_{k-1}\right].$$

Summing the above relation for all $n \in \{0\}\cup[\frac{H}{h}-1]$, using linearity of expectation, and the fact $\bar{V}_{H+1}^k(s) = $ for all $s,k$,

$$\sum_{n=0}^{\frac{H}{h}-1}\mathbb{E}\left[\bar{V}_{1+nh}^k(s_t^k)\mid\mathcal{F}_{k-1}\right] \leq \frac{H}{h}\epsilon_V + \mathbb{E}\left[\sum_{t=1}^{H}r(s_t^k,a_t^k)\mid\mathcal{F}_{k-1}\right] + \sum_{n=1}^{\frac{H}{h}-1}\mathbb{E}\left[\bar{V}_{1+nh}^{k-1}(s_{1+nh}^k)\mid\mathcal{F}_{k-1}\right]$$

$$\iff \bar{V}_1^k(s_1^k) + \sum_{n=1}^{\frac{H}{h}-1}\mathbb{E}\left[\bar{V}_{1+nh}^k(s_t^k)\mid\mathcal{F}_{k-1}\right] \leq \frac{H}{h}\epsilon_V + \mathbb{E}\left[\sum_{t=1}^{H}r(s_t^k,a_t^k)\mid\mathcal{F}_{k-1}\right] + \sum_{n=1}^{\frac{H}{h}-1}\mathbb{E}\left[\bar{V}_{1+nh}^{k-1}(s_{1+nh}^k)\mid\mathcal{F}_{k-1}\right]$$

$$\iff \bar{V}_1^k(s_1^k) + \sum_{n=1}^{\frac{H}{h}-1}\mathbb{E}\left[\bar{V}_{1+nh}^k(s_t^k)\mid\mathcal{F}_{k-1}\right] \leq \frac{H}{h}\epsilon_V + V^{\pi_k}(s_1^k) + \sum_{n=1}^{\frac{H}{h}-1}\mathbb{E}\left[\bar{V}_{1+nh}^{k-1}(s_{1+nh}^k)\mid\mathcal{F}_{k-1}\right]$$

$$\iff \bar{V}_1^k(s_1^k) - V^{\pi_k}(s_1^k) \leq \frac{H}{h}\epsilon_V + \sum_{n=1}^{\frac{H}{h}-1}\mathbb{E}\left[\bar{V}_{1+nh}^{k-1}(s_{1+nh}^k) - \bar{V}_{1+nh}^k(s_{1+nh}^k)\mid\mathcal{F}_{k-1}\right]$$

$$\iff \bar{V}_1^k(s_1^k) - V^{\pi_k}(s_1^k) \leq \frac{H}{h}\epsilon_V + \sum_{k=1}^{K}\sum_{n=1}^{\frac{H}{h}-1}\sum_{s}\bar{V}_{nh+1}^{k-1}(s) - \mathbb{E}[\bar{V}_{nh+1}^k(s)\mid\mathcal{F}_{k-1}]$$

The second line holds by the fact $s_1^k$ is measurable w.r.t. $\mathcal{F}_{k-1}$, and the third line holds since

$$V_1^{\pi_k}(s_1^k) = \mathbb{E}\left[\sum_{t=1}^{H}r(s_t^k,a_t^k)\mid\mathcal{F}_{k-1}\right].$$

The fifth line holds by by Lemma 15 with $\bar{V}_t^k = g_t^k$ for $t = nh+1$. See that the update of $\bar{V}_t^k$ occurs only at the visited state $s_t^k$ and the update rule uses $\bar{V}_{t+1}^{k-1}$, i.e., it is measurable w.r.t. to $\mathcal{F}_{k-1}$, and it is valid to apply the lemma.

$\square$

**Theorem 6** (Performance of $h$-RTDP-AV). *Let $\epsilon,\delta > 0$. The following holds for $h$-RTDP-AV:*

1. *With probability $1-\delta$, for all $K > 0$,* $\operatorname{Regret}(K) \leq \frac{9SH(H-h)}{h}(1+\frac{H}{h}\epsilon_V)\ln(\frac{3}{\delta}) + \frac{2H}{h}\epsilon_V K.$

2. *Let $\Delta_V = 2H\epsilon_V$. Then,* $\Pr\left\{\exists\epsilon > 0 : N_\epsilon^{\frac{\Delta_V}{h}} \geq \frac{9SH(H-h)(1+\frac{\Delta_V}{2h})\ln(\frac{3}{\delta})}{h\epsilon}\right\} \leq \delta.$

*Proof.* We start by proving **claim (1)**. The following bounds on the regret hold.

$$\text{Regret}(K) := \sum_{k=1}^{K} V_1^*(s_1^k) - V_1^{\pi_k}(s_1^k)$$

$$\leq \sum_{k=1}^{K} \bar{V}_1^k(s_1^k) - V_1^{\pi_k}(s_1^k) + \frac{H}{h}\epsilon_V$$

$$= \frac{2H}{h}\epsilon_V K + \sum_{k=1}^{K} \sum_{n=1}^{\frac{H}{h}-1} \sum_s \bar{V}_{nh+1}^{k-1}(s) - \mathbb{E}[\bar{V}_{nh+1}^k(s) \mid \mathcal{F}_{k-1}] \qquad (27)$$

The second relation is by the approximated optimism of the value function when approximate value updates are used (Lemma 11). The third relation is by Lemma 12.

We now observe the regret is a regret of a Decreasing Bounded Process. Let

$$X_k := \sum_{n=1}^{\frac{H}{h}-1} \sum_s \bar{V}_{nh+1}^k(s), \qquad (28)$$

and observe that $\{X_k\}_{g \geq 0}$ is a Decreasing Bounded Process.

1. It is decreasing since for all $s, t$ $\bar{V}_t^k(s) \leq \bar{V}_t^{k-1}(s)$ by Lemma 11. Thus, their sum is also decreasing.

2. It is bounded since for all $s, n \in [\frac{H}{h}] - 1$,

$$\bar{V}_{1+hn}^k(s) \geq V_{1+hn}^*(s) - \epsilon_V\left(\frac{H}{h} - n\right) \geq -\epsilon_V\left(\frac{H}{h} - n\right) \geq -\epsilon_V\frac{H}{h}$$

by Lemma 11. Thus, $X_0$ which is a sum of the above terms is bounded from below by $-\frac{\epsilon_V}{h}\frac{SH(H-h)}{h}$.

See that the initial value can be bounded as follows,

$$X_0 = \sum_{n=1}^{\frac{H}{h}-1} \sum_s \bar{V}_{nh+1}^0(s) \leq \sum_{n=1}^{\frac{H}{h}-1} \sum_s H = \frac{SH(H-h)}{h}.$$

Using linearity of expectation and the definition (14) we observe that (27) can be written,

$$\text{Regret}(K) \leq (27) = \frac{2H}{h}\epsilon_V K + \sum_{k=1}^{K} X_{k-1} - \mathbb{E}[X_k \mid \mathcal{F}_{k-1}],$$

which is regret of A Bounded Decreasing Process. Applying the regret bound on DBP, Theorem 1 we conclude the proof of the first claim.

We now prove **claim (2)** using the proving technique at Theorem 4. Denote $\Delta_V = 2H\epsilon_V$. The following relations hold for all $\epsilon > 0$.

$$\mathbb{1}\left\{\bar{V}_1^k(s_1^k) - V_1^{\pi_k}(s_1^k) \geq \frac{\Delta_V}{2h} + \epsilon\right\}\left(\epsilon + \frac{\Delta_V}{2h}\right)$$

$$\leq \mathbb{1}\left\{\bar{V}_1^k(s_1^k) - V_1^{\pi_k}(s_1^k) \geq \frac{\Delta_V}{2h} + \epsilon\right\}(\bar{V}_1^k(s_1^k) - V_1^{\pi_k}(s_1^k))$$

$$= \mathbb{1}\left\{\bar{V}_1^k(s_1^k) - V_1^{\pi_k}(s_1^k) \geq \frac{\Delta_V}{2h} + \epsilon\right\}\left(\sum_{n=1}^{\frac{H}{h}-1} \sum_s \bar{V}_{nh+1}^{k-1}(s) - \mathbb{E}[\bar{V}_{nh+1}^k(s) \mid \mathcal{F}_{k-1}] + \frac{\Delta_V}{2h}\right)$$

$$= \mathbb{1}\left\{\bar{V}_1^k(s_1^k) - V_1^{\pi_k}(s_1^k) \geq \frac{\Delta_V}{2h} + \epsilon\right\}\left(X_{k-1} - \mathbb{E}[X_k \mid \mathcal{F}_{k-1}] + \frac{\Delta_V}{2h}\right).$$

The first relation holds by the indicator function and the second relation by Lemma 12. The third relation holds by the definition of $X_k$ (28) and linearity of expectation. Using an algebraic manipulation the above leads to the following relation,

$$\mathbb{1}\left\{\bar{V}_1^k(s_1^k) - V_1^{\pi_k}(s_1^k) \geq \frac{\Delta_V}{2h} + \epsilon\right\}\epsilon \leq \mathbb{1}\left\{\bar{V}_1^k(s_1^k) - V_1^{\pi_k}(s_1^k) \geq \frac{\Delta_V}{2h} + \epsilon\right\}(X_{k-1} - \mathbb{E}[X_k \mid \mathcal{F}_{k-1}])$$
(29)

As we wish the final performance to be compared to $V^*$ we use the the first claim of Lemma 11, by which for all $s, k$, $\bar{V}_1^k(s) \geq V_1^*(s) - \frac{\Delta_V}{2h}$. This implies that

$$\mathbb{1}\left\{V_1^*(s_1^k) - V_1^{\pi_k}(s_1^k) \geq \frac{\Delta_V}{h} + \epsilon\right\} \leq \mathbb{1}\left\{\bar{V}_1^k(s_1^k) - V_1^{\pi_k}(s_1^k) \geq \frac{\Delta_V}{2h} + \epsilon\right\}.$$
(30)

Combining the above we get

$$\mathbb{1}\left\{V_1^*(s_1^k) - V_1^{\pi_k}(s_1^k) \geq \frac{\Delta_V}{h} + \epsilon\right\}\epsilon$$

$$\leq \mathbb{1}\left\{\bar{V}_1^k(s_1^k) - V_1^{\pi_k}(s_1^k) \geq \frac{\Delta_V}{2h} + \epsilon\right\}\epsilon$$

$$\leq \mathbb{1}\left\{\bar{V}_1^k(s_1^k) - V_1^{\pi_k}(s_1^k) \geq \frac{\Delta_V}{2h} + \epsilon\right\}(X_{k-1} - \mathbb{E}[X_k \mid \mathcal{F}_{k-1}]).$$
(31)

The first relation is by (30) and the second relation by (29).

Define $N_\epsilon(K) = \sum_{k=1}^K \mathbb{1}\left\{V_1^*(s_1^k) - V_1^{\pi_k}(s_1^k) \geq \frac{\Delta_V}{h} + \epsilon\right\}$ as the number of times $V_1^*(s_1^k) - V_1^{\pi_k}(s_1^k) \geq \frac{\Delta_V}{h} + \epsilon$ at the first $K$ episodes. Summing the above inequality (31) for all $k \in [K]$ and denote we get that for all $\epsilon > 0$

$$N_\epsilon(K)\epsilon = \sum_{k=1}^K \mathbb{1}\left\{V_1^*(s_1^k) - V_1^{\pi_k}(s_1^k) \geq \frac{\Delta_V}{h} + \epsilon\right\}\epsilon$$

$$\leq \sum_{k=1}^K \mathbb{1}\left\{\bar{V}_1^k(s_1^k) - V_1^{\pi_k}(s_1^k) \geq \frac{\Delta_V}{2h} + \epsilon\right\}(X_{k-1} - \mathbb{E}[X_k \mid \mathcal{F}_{k-1}])$$

$$\leq \sum_{k=1}^K X_{k-1} - \mathbb{E}[X_k \mid \mathcal{F}_{k-1}].$$

The first relation holds by definition, the second by (31) and the third relation holds as $\{X_k\}_{k\geq 0}$ is a DBP (28) and, thus, $X_{k-1} - \mathbb{E}[X_k \mid \mathcal{F}_{k-1}] \geq 0$ a.s. . Thus, the following relation holds

$$\left\{\forall K > 0 : \sum_{k=1}^K X_{k-1} - \mathbb{E}[X_k \mid \mathcal{F}_{k-1}] \leq \frac{9SH(H-h)}{h}(1 + \frac{H}{h}\epsilon_V)\ln\frac{3}{\delta}\right\}$$

$$\subseteq \left\{\forall \epsilon > 0 : N_\epsilon(K)\epsilon \leq \frac{9SH(H-h)}{h}(1 + \frac{H}{h}\epsilon_V)\ln\frac{3}{\delta}\right\},$$

from which we get for any $K > 0$

$$\Pr\left(\forall \epsilon > 0 : N_\epsilon(K)\epsilon \leq \frac{9SH(H-h)}{h}(1 + \frac{H}{h}\epsilon_V)\ln\frac{3}{\delta}\right)$$

$$\geq \Pr\left(\forall K > 0 : \sum_{k=1}^K X_{k-1} - \mathbb{E}[X_k \mid \mathcal{F}_{k-1}] \leq \frac{9SH(Hh)}{h}(1 + \frac{H}{h}\epsilon_V)\ln\frac{3}{\delta}\right) \geq 1 - \delta,$$

and the third relation holds the bound on the regret of DBP, Theorem 1. Equivalently, for any $K > 0$,

$$\Pr\left(\exists \epsilon > 0 : N_\epsilon(K)\epsilon \geq \frac{9SH(H-h)}{h}(1 + \frac{H}{h}\epsilon_V)\ln\frac{3}{\delta}\right) \leq \delta.$$
(32)

Applying the Monotone Convergence Theorem as in the proof of Theorem 4 we conclude the proof.

$\square$

**Algorithm 8** $h$-RTDP with Approximate State Abstraction ($h$-RTDP-AA)

---

init: $\forall s_\phi \in \mathcal{S}_\phi, \ n \in \{0\} \cup [\frac{H}{h}], \ \bar{V}^0_{\phi,nh+1}(s_\phi) = H - nh$

**for** $k \in [K]$ **do**

    Initialize $s_1^k$

    **for** $t \in [H]$ **do**

        **if** $(t-1) \mod h == 0$ **then**

            $h_c = t + h \ ; \qquad \bar{V}^k_{\phi,t}(\phi_t(s_t^k)) = T^h_\phi \bar{V}^{k-1}_{\phi,h_c}(s_t^k) \ ;$

            $\bar{V}^k_{\phi,t}(\phi_t(s_t^k)) \leftarrow \min\left\{ \bar{V}^k_{\phi,t}(\phi_t(s_t^k)), \bar{V}^{k-1}_{\phi,t}(\phi_t(s_t^k)) \right\} \ ;$

        **end if**

        $a_t^k \in \arg\max_a r(s_t^k, a) + p(\cdot | s_t^k, a) T^{h_c - t - 1}_\phi \bar{V}^{k-1}_{\phi,h_c} \ ;$

        Act with $a_t^k$ and observe $s_{t+1}^k \sim p(\cdot | s_t^k, a_t^k)$

    **end for**

**end for**

---

## 13   $h$-RTDP with Approximate State Abstraction

In this section we analyze the performance of $h$-RTDP performance which uses approximate abstraction. For clarity we restate the assumption we make on the approximate abstraction and the definition of equivalent set under abstraction.

**Assumption 1** (Approximate Abstraction, [23], definition 3.3). *For any $s, s' \in \mathcal{S}$ and $n \in \{0\} \cup [\frac{H}{h} - 1]$ for which $\phi_{nh+1}(s) = \phi_{nh+1}(s')$, we have $|V^*_{nh+1}(s) - V^*_{nh+1}(s')| \leq \epsilon_A$.*

An important quantity in our analysis is the set of states equivalent to a given state $s$ under $\phi_{nh+1}$.

**Definition 1** (Equivalent Set Under Abstraction). *For any $s \in \mathcal{S}$ and $n \in \{0\} \cup [\frac{H}{h} - 1]$, we define the set of states equivalent to $s$ under $\phi_{nh+1}$ as $\Phi_{nh+1}(s) := \{s' \in \mathcal{S} : \phi_{nh+1}(s) = \phi_{nh+1}(s')\}$.*

Before we supply with the proof we emphasize an important difference in the definition of the value function $\bar{V}_t^k$ when using abstraction. Unlike the usual definition of $\bar{V}_t^k : \mathcal{S} \to \mathbb{R}$, in case of abstraction $\bar{V}^k_{\phi,t}$ is a mapping from the *abstract state space* to the reals, i.e., $\bar{V}^k_{\phi,t} : \mathcal{S}_\phi \to \mathbb{R}$. Meaning, $\bar{V}^k_{\phi,t}$ is defined on the abstract state space. Given a state $s \in \mathcal{S}$ we need to query $\phi_t$ to obtain its value at time $t$ by $\bar{V}^k_t(\phi_t(s))$.

**Lemma 13.** *For all $s \in \mathcal{S}$, $n \in \{0\} \cup [\frac{H}{h}]$, and $k \in [K]$:*

  *(i) Optimism:*

$$\max_{s' \in \Phi_{nh+1}(s)} V^*_{nh+1}(s') \leq \bar{V}^k_{nh+1}(\phi_{nh+1}(s)) + \epsilon_A(\frac{H}{h} - n).$$

  *(ii) Bounded: $\bar{V}^k_{nh+1}(\phi_{nh+1}(s)) \geq 0$.*

  *(iii) Non-Increasing: $\bar{V}^k_{nh+1}(\phi_{nh+1}(s)) \leq \bar{V}^{k-1}_{nh+1}(\phi_{nh+1}(s))$.*

*Proof.* We prove the first claim by induction. The second and third claims hold by construction.

*(i)* Let $n \in \{0\} \cup [\frac{H}{h} - 1]$. By the optimistic initialization, $\forall s, n, \ V^*_{1+hn}(s) - \epsilon_A(\frac{H}{h} - n) \leq V^*_{1+hn}(s) \leq V^0_{1+hn}(\phi_{1+hn}(s))$. Assume the claim holds for $k - 1$ episodes. Let $s_t^k$ be the state the algorithm is at in the $t = 1 + hn$ time step of the $k$'th episode, i.e., at a time step in which a value update is taking place. By the value update of Algorithm 8,

$$\bar{V}^k_t(\phi(s_t^k)) = \min\{T^h \bar{V}^{k-1}_{h_c}(s_t^k), \bar{V}^{k-1}_t(\phi(s_t^k))\}. \tag{33}$$

If the minimal value is $\bar{V}^{k-1}_t(\phi(s_t^k))$ then $\bar{V}^k_t(\phi(s_t^k))$ satisfies the induction hypothesis by the induction assumption. If $T^h \bar{V}^{k-1}_{h_c}(s_t^k)$ is the minimal value in (33), then the following relation holds,

$$\bar{V}_t^k(\phi_t(s_t^k)) = \max_{\pi_0,\pi_1,..,\pi_{h-1}} \mathbb{E}[\sum_{t'=0}^{h-1} r(s_t', \pi_{t'}(s_t')) + \bar{V}_{t+h}^{k-1}(\phi(s_h)) \mid s_0 = s_t^k]$$

$$\geq \max_{\pi_0,\pi_1,..,\pi_{h-1}} \mathbb{E}[\sum_{t'=0}^{h-1} r(s_t', \pi_{t'}(s_t')) + \max_{s' \in \Phi_{t+h}(s_h)} V_{t+h}^*(s') - \epsilon_A \left(\frac{H}{h} - n - 1\right) \mid s_0 = s_t^k]$$

$$= \max_{\pi_0,\pi_1,..,\pi_{h-1}} \mathbb{E}[\sum_{t'=0}^{h-1} r(s_t', \pi_{t'}(s_t')) + \max_{s' \in \Phi_{t+h}(s_h)} V_{t+h}^*(s') \mid s_0 = s_t^k] - \epsilon_A \left(\frac{H}{h} - n - 1\right)$$

$$\geq \max_{\pi_0,\pi_1,..,\pi_{h-1}} \mathbb{E}[\sum_{t'=0}^{h-1} r(s_t', \pi_{t'}(s_t')) + V_{t+h}^*(s_h) \mid s_0 = s_t^k] - \epsilon_A \left(\frac{H}{h} - n - 1\right)$$

$$= V_t^*(s_t^k) - \epsilon_A \left(\frac{H}{h} - n - 1\right)$$

$$\geq \max_{s' \in \Phi_t(s_t^k)} V_t^*(s_t^k) - \epsilon_A - \epsilon_A \left(\frac{H}{h} - n - 1\right)$$

$$= \max_{s' \in \Phi_t(s_t^k)} V_t^*(s_t^k) - \epsilon_A \left(\frac{H}{h} - n\right).$$

The first relation is the definition of the update rule. The second relation holds by the monotonicity of the max operator together with the induction assumption. The third relation as the extracted term out of the max is constant. The forth relation holds by the definition of the max operation. The fifth relation by the Bellman equations $V_t^*$ satisfies (2), and the sixth relation by Assumption 1.

**(ii)** The second claim holds by construction of the update rule $\bar{V}_t^k(s_t^k) \leftarrow \min\{\bar{V}_t^k(s_t^k), \bar{V}_t^{k-1}(s_t^k)\}$ which enforces $\bar{V}_t^k(s) \leq \bar{V}_t^{k-1}(s)$ for every updated state, and thus for all $s$ and $t$.

**(iii)** The third claim holds since $V_t^k(\phi_t(s))$ is initialized with positive elements and is updated by itself and positive elements, as $r(s, a) \geq 0$. Thus, it remains positive a.s. . $\qquad\square$

**Lemma 14.** *The expected cumulative value update at the k'th episode of h-RTDP-AA satisfies the following relation:*

$$\bar{V}_1^k(\phi(s_1^k)) - V_1^{\pi_k}(s_1^k)$$

$$\leq \sum_{k=1}^K \sum_{n=1}^{\frac{H}{h}-1} \sum_{s_\phi \in \mathcal{S}_\phi} \bar{V}_{nh+1}^{k-1}(s_\phi) - \mathbb{E}[\bar{V}_{nh+1}^k(s_\phi) \mid \mathcal{F}_{k-1}].$$

*Proof.* Let $n \in \{0\} \cup [\frac{H}{h} - 1]$ and let $t = 1 + hn$ be a time step in which a value update is taking place. By the definition of the update rule, the following holds for the update at the visited state $s_t^k$:

$$\bar{V}_t^k(\phi_t(s_t^k)) \leq \mathbb{E}\left[\sum_{t'=t}^{t+h-1} r(s_{t'}^k, a_{t'}^k) + \bar{V}_{t+h}^{k-1}(\phi_{t+h}(s_{t+h}^k)) \mid \pi_k, s_t^k\right]$$

$$= \mathbb{E}\left[\sum_{t'=t}^{t+h-1} r(s_{t'}^k, a_{t'}^k) + \bar{V}_{t+h}^{k-1}(\phi_{t+h}(s_{t+h}^k)) \mid \mathcal{F}_{k-1}, s_t^k\right]$$

where the last relation follows by the same argument as in (10).

Taking the conditional expectation w.r.t. $\mathcal{F}_{k-1}$ and using the tower property we get,

$$\mathbb{E}\left[\bar{V}_t^k(\phi_t(s_t^k)) \mid \mathcal{F}_{k-1}\right] \leq \mathbb{E}\left[\sum_{t'=t}^{t+h-1} r(s_{t'}^k, a_{t'}^k) + \bar{V}_{t+h}^{k-1}(\phi_{t+h}(s_{t+h}^k)) \mid \mathcal{F}_{k-1}\right].$$

Denote $s_{\phi,t}^k := \phi_t(s_t^k)$. Summing the above relation for all $n \in \{0\} \cup [\frac{H}{h} - 1]$, using linearity of expectation, and the fact $\bar{V}_{H+1}^k(\phi_{H+1}(s)) = 0$ for all $s, k$,

$$\sum_{n=0}^{\frac{H}{h}-1} \mathbb{E}\big[\bar{V}_{1+nh}^k(s_{\phi,1+nh}^k) \mid \mathcal{F}_{k-1}\big] \le \mathbb{E}\Big[\sum_{t=1}^{H} r(s_t^k, a_t^k) \mid \mathcal{F}_{k-1}\Big] + \sum_{n=1}^{\frac{H}{h}-1} \mathbb{E}\big[\bar{V}_{1+nh}^{k-1}(s_{\phi,1+nh}^k) \mid \mathcal{F}_{k-1}\big]$$

$$\Longleftrightarrow \bar{V}_1^k(s_{\phi,1}^k) + \sum_{n=1}^{\frac{H}{h}-1} \mathbb{E}\big[\bar{V}_{1+nh}^k(s_{\phi,1+nh}^k) \mid \mathcal{F}_{k-1}\big] \le \mathbb{E}\Big[\sum_{t=1}^{H} r(s_t^k, a_t^k) \mid \mathcal{F}_{k-1}\Big] + \sum_{n=1}^{\frac{H}{h}-1} \mathbb{E}\big[\bar{V}_{1+nh}^{k-1}(s_{\phi,1+nh}^k) \mid \mathcal{F}_{k-1}\big]$$

$$\Longleftrightarrow \bar{V}_1^k(s_{\phi,1}^k) + \sum_{n=1}^{\frac{H}{h}-1} \mathbb{E}\big[\bar{V}_{1+nh}^k(s_{\phi,1+nh}^k) \mid \mathcal{F}_{k-1}\big] \le V^{\pi_k}(s_1^k) + \sum_{n=1}^{\frac{H}{h}-1} \mathbb{E}\big[\bar{V}_{1+nh}^{k-1}(s_{\phi,1+nh}^k) \mid \mathcal{F}_{k-1}\big]$$

$$\Longleftrightarrow \bar{V}_1^k(s_{\phi,1}^k) - V^{\pi_k}(s_1^k) \le \sum_{n=1}^{\frac{H}{h}-1} \mathbb{E}\big[\bar{V}_{1+nh}^{k-1}(s_{\phi,1+nh}^k) - \bar{V}_{1+nh}^k(s_{\phi,1+nh}^k) \mid \mathcal{F}_{k-1}\big]$$

$$\Longleftrightarrow \bar{V}_1^k(s_{\phi,1}^k) - V^{\pi_k}(s_1^k) \le \sum_{k=1}^{K} \sum_{n=1}^{\frac{H}{h}-1} \sum_{s_\phi \in \mathcal{S}_\phi} \bar{V}_{nh+1}^{k-1}(s_\phi) - \mathbb{E}\big[\bar{V}_{nh+1}^k(s_\phi) \mid \mathcal{F}_{k-1}\big]$$

The second line holds by the fact $s_1^k$ is measurable w.r.t. $\mathcal{F}_{k-1}$, the third line holds since

$$V_1^{\pi_k}(s_1^k) = \mathbb{E}\Big[\sum_{t=1}^{H} r(s_t^k, a_t^k) \mid \mathcal{F}_{k-1}\Big].$$

The fifth line holds by Lemma 15 with $\bar{V}_t^k = g_t^k$ for $t = nh + 1$. Furthermore, we set $\tilde{\mathcal{S}}$ of Lemma 15 to be $\mathcal{S}_\phi$. See that the update of $\bar{V}_t^k$ occurs only at the visited state $s_{\phi,t}^k = \phi(s_t^k)$ of the abstracted state space. Furthermore, the update rule uses $\bar{V}_{\phi,t+1}^{k-1}$, i.e., it is measurable w.r.t. to $\mathcal{F}_{k-1}$, and it is valid to apply the lemma. $\qquad\square$

**Theorem 7** (Performance of $h$-RTDP-AA). *Let $\epsilon, \delta > 0$. The following holds for $h$-RTDP-AA:*

1. *With probability $1 - \delta$, for all $K > 0$,* $\quad \mathrm{Regret}(K) \le \frac{9S_\phi H(H-h)}{h} \ln(3/\delta) + \frac{H\epsilon_A}{h} K.$

2. *Let $\Delta_A = H\epsilon_A$. Then,* $\quad \Pr\left\{\exists \epsilon > 0 \; : \; N_\epsilon^{\frac{\Delta_A}{h}} \ge \frac{9S_\phi H(H-h) \ln(3/\delta)}{h\epsilon}\right\} \le \delta.$

Before supplying with the proof observe the following remark.

*Proof.* We start by proving **claim (1)**. The following bounds on the regret hold.

$$\mathrm{Regret}(K) := \sum_{k=1}^{K} V_1^*(s_1^k) - V_1^{\pi_k}(s_1^k)$$

$$\le \sum_{k=1}^{K} \max_{s \in \Phi_1(s_1^k)} V_1^*(s) - V_1^{\pi_k}(s_1^k)$$

$$\le \sum_{k=1}^{K} \bar{V}_1^k(\phi_1(s_1^k)) - V_1^{\pi_k}(s_1^k) + \epsilon_A \frac{H}{h}$$

$$\le \epsilon_A \frac{H}{h} K + \sum_{k=1}^{K} \sum_{n=1}^{\frac{H}{h}-1} \sum_{s_\phi \in \mathcal{S}_\phi} \bar{V}_{nh+1}^{k-1}(s_\phi) - \mathbb{E}\big[\bar{V}_{nh+1}^k(s_\phi) \mid \mathcal{F}_{k-1}\big] \qquad (34)$$

The second relation holds the definition of the $\max$ operator and since $s_1^k \in \phi(s_1^k)$ (by definition we have that $s \in \Phi_t(s)$, as $\phi_t(s) = \phi_t(s)$ for any $t$). The third relation holds by the approximate optimism of the value function (Lemma 13), and the forth relation is by Lemma 14.

We now observe the regret is a regret of a Decreasing Bounded Process. Let

$$X_k := \sum_{n=1}^{\frac{H}{h}-1} \sum_{s_\phi \in \mathcal{S}_\phi} \bar{V}_{nh+1}^k(s_\phi), \tag{35}$$

and observe that $\{X_k\}_{g \geq 0}$ is a Decreasing Bounded Process.

1. It is decreasing since for all $s_\phi \in \mathcal{S}_\phi, t\ \bar{V}_t^k(s_\phi) \leq \bar{V}_t^{k-1}(s_\phi)$ by Lemma 13. Thus, their sum is also decreasing.

2. It is bounded since for all $s \in \mathcal{S}_\phi, t\ \bar{V}_t^k(s_\phi) \geq 0$ by Lemma 13. Thus, the sum is bounded from below by 0.

See that the initial value can be bounded as follows,

$$X_0 = \sum_{n=1}^{\frac{H}{h}-1} \sum_{s_\phi \in \mathcal{S}_\phi} \bar{V}_{nh+1}^0(s_\phi) \leq \sum_{n=1}^{\frac{H}{h}-1} \sum_{s_\phi \in \mathcal{S}_\phi} H = \frac{S_\phi H(H-h)}{h}.$$

Using linearity of expectation and the definition (14) we observe that (34) can be written,

$$\mathrm{Regret}(K) \leq (34) = \epsilon_A \frac{H}{h} K + \sum_{k=1}^{K} X_{k-1} - \mathbb{E}[X_k \mid \mathcal{F}_{k-1}],$$

which is regret of A Bounded Decreasing Process. Applying the bound on the regret of a DRP, Theorem 1, we conclude the proof of the first claim.

We now prove **claim (2)** using the proving technique at Theorem 4. Denote $\Delta_A = H\epsilon_A$. The following relations hold for all $\epsilon > 0$.

$$\mathbb{1}\{\bar{V}_1^k(\phi_1(s_1^k)) - V_1^{\pi_k}(s_1^k) \geq \epsilon\}\epsilon$$
$$\leq \mathbb{1}\{\bar{V}_1^k(\phi_1(s_1^k)) - V_1^{\pi_k}(s_1^k) \geq \epsilon\}(\bar{V}_1^k(\phi_1(s_1^k)) - V_1^{\pi_k}(s_1^k))$$
$$\leq \mathbb{1}\{\bar{V}_1^k(\phi_1(s_1^k)) - V_1^{\pi_k}(s_1^k) \geq \epsilon\}\left(\sum_{n=1}^{\frac{H}{h}-1} \sum_{s_\phi \in \mathcal{S}_\phi} \bar{V}_{nh+1}^{k-1}(s_\phi) - \mathbb{E}[\bar{V}_{nh+1}^k(s_\phi) \mid \mathcal{F}_{k-1}]\right)$$
$$= \mathbb{1}\{\bar{V}_1^k(s_1^k) - V_1^{\pi_k}(s_1^k) \geq \epsilon\}(X_{k-1} - \mathbb{E}[X_k \mid \mathcal{F}_{k-1}]). \tag{36}$$

The first relation holds by the indicator function and the second relation holds by Lemma 14. The forth relation holds by the definition of $X_k$ (35) and linearity of expectation.

As we wish the final performance to be compared to $V^*$ we use the the first claim of Lemma 13, by which for all $s, k, \bar{V}_1^k(\phi_1(s)) \geq V_1^*(s) - \frac{\Delta_A}{h}$. This implies that

$$\mathbb{1}\left\{V_1^*(s_1^k) - V_1^{\pi_k}(s_1^k) \geq \frac{\Delta_A}{h} + \epsilon\right\} \leq \mathbb{1}\{\bar{V}_1^k(\phi_1(s_1^k)) - V_1^{\pi_k}(s_1^k) \geq \epsilon\}. \tag{37}$$

Combining the above we get

$$\mathbb{1}\left\{V_1^*(s_1^k) - V_1^{\pi_k}(s_1^k) \geq \frac{\Delta_A}{h} + \epsilon\right\}\epsilon$$
$$\leq \mathbb{1}\{\bar{V}_1^k(\phi_1(s_1^k)) - V_1^{\pi_k}(s_1^k) \geq \epsilon\}\epsilon$$
$$\leq \mathbb{1}\{\bar{V}_1^k(\phi_1(s_1^k)) - V_1^{\pi_k}(s_1^k) \geq \epsilon\}(X_{k-1} - \mathbb{E}[X_k \mid \mathcal{F}_{k-1}]). \tag{38}$$

The first relation is by (37) and the second relation by (36).

Define $N_\epsilon(K) = \sum_{k=1}^{K} \mathbb{1}\{V_1^*(s_1^k) - V_1^{\pi_k}(s_1^k) \geq \frac{\Delta_A}{h} + \epsilon\}$ as the number of times $V_1^*(s_1^k) - V_1^{\pi_k}(s_1^k) \geq \frac{\Delta_A}{h} + \epsilon$ at the first $K$ episodes. Summing the above inequality (38) for all $k \in [K]$ and

denote we get that for all $\epsilon > 0$

$$N_\epsilon(K)\epsilon = \sum_{k=1}^K \mathbb{1}\left\{V_1^*(s_1^k) - V_1^{\pi_k}(s_1^k) \geq \frac{\Delta_A}{h} + \epsilon\right\}\epsilon$$

$$\leq \sum_{k=1}^K \mathbb{1}\left\{\bar{V}_1^k(\phi_1(s_1^k)) - V_1^{\pi_k}(s_1^k) \geq \epsilon\right\}(X_{k-1} - \mathbb{E}[X_k \mid \mathcal{F}_{k-1}])$$

$$\leq \sum_{k=1}^K X_{k-1} - \mathbb{E}[X_k \mid \mathcal{F}_{k-1}].$$

The first relation holds by definition, the second by (38) and the third relation holds as $\{X_k\}_{k \geq 0}$ is a DBP (35) and, thus, $X_{k-1} - \mathbb{E}[X_k \mid \mathcal{F}_{k-1}] \geq 0$ a.s. . Thus, the following relation holds

$$\left\{\forall K > 0 : \sum_{k=1}^K X_{k-1} - \mathbb{E}[X_k \mid \mathcal{F}_{k-1}] \leq \frac{9SH(H-h)}{h}\ln\frac{3}{\delta}\right\} \subseteq \left\{\forall \epsilon > 0 : N_\epsilon(K)\epsilon \leq \frac{9SH(H-h)}{h}\ln\frac{3}{\delta}\right\},$$

from which we get that for any $K > 0$

$$\Pr\left(\forall \epsilon > 0 : N_\epsilon(K)\epsilon \leq \frac{9SH(H-h)}{h}\ln\frac{3}{\delta}\right)$$

$$\geq \Pr\left(\forall K > 0 : \sum_{k=1}^K X_{k-1} - \mathbb{E}[X_k \mid \mathcal{F}_{k-1}] \leq \frac{9SH(Hh)}{h}\ln\frac{3}{\delta}\right) \geq 1 - \delta,$$

and the third relation holds the bound on the regret of DBP, Theorem 1. Equivalently, for any $K > 0$,

$$\Pr\left(\exists \epsilon > 0 : N_\epsilon(K)\epsilon \geq \frac{9SH(H-h)}{h}\ln\frac{3}{\delta}\right) \leq \delta. \tag{39}$$

Applying the Monotone Convergence Theorem as in the proof of Theorem 4 we conclude the proof.

$\square$

## 14 Useful Lemmas

The following lemma is a generalization of Lemma 34 in [15].

**Lemma 15** (On Trajectory Regret to Uniform Regret)**.** *For any $t \in [H]$, let $\left\{ s_t^k, \mathcal{F}_k \right\}_{k \geq 0}$ be a random process where $\left\{ s_t^k \right\}_{k \geq 0}$ is adapted to the filtration $\{\mathcal{F}_k\}_{k \geq 0}$ and $s_t^k \in \tilde{\mathcal{S}}$ where $\tilde{\mathcal{S}}$ is a finite set of all possible realizations of $s_t^k$ with cardinally $\tilde{S} := |\tilde{\mathcal{S}}|$. Let $g_t^k \in \mathbb{R}^{\tilde{S}}$ and denoting the $s \in \tilde{\mathcal{S}}$ entry of the vector as $g_t^k(s)$. Furthermore, let $g_t^k(s)$ be updated only at the state $s_t^k$ by an update rule which is $\mathcal{F}_{k-1}$ measurable, i.e.,*

$$g_t^k(s) = \begin{cases} f_t^{k-1}(s), \text{if } s = s_t^k, \\ g_t^{k-1}(s), \text{o.w..} \end{cases}$$

*Where $f_t^{k-1}(s)$ is an update rule $\mathcal{F}_{k-1}$ measurable. Then,*

$$\sum_{k=1}^{K} \mathbb{E}[g_t^{k-1}(s_t^k) - g_t^k(s_t^k) \mid \mathcal{F}_{k-1}] = \sum_{k=1}^{K} \sum_{s \in \tilde{\mathcal{S}}} g_t^{k-1}(s) - \mathbb{E}[g_t^k(s) \mid \mathcal{F}_{k-1}]$$

*Proof.* The following relations hold.

$$\sum_{k=1}^{K} \sum_{t=1}^{H} \mathbb{E}[g_t^{k-1}(s_t^k) - g_t^k(s_t^k) \mid \mathcal{F}_{k-1}]$$

$$= \sum_{k=1}^{K} \sum_{t=1}^{H} \sum_{s \in \tilde{\mathcal{S}}} \mathbb{E}[\mathbb{1}\{s = s_t^k\} g_t^{k-1}(s) - \mathbb{1}\{s = s_t^k\} g_t^k(s) \mid \mathcal{F}_{k-1}]$$

$$\overset{(1)}{=} \sum_{k=1}^{K} \sum_{t=1}^{H} \sum_{s \in \tilde{\mathcal{S}}} \mathbb{E}[\mathbb{1}\{s = s_t^k\} g_t^{k-1}(s) - \mathbb{1}\{s = s_t^k\} f_t^{k-1}(s) \mid \mathcal{F}_{k-1}]$$

$$\overset{(2)}{=} \sum_{t=1}^{H} \sum_{s \in \tilde{\mathcal{S}}} \sum_{k=1}^{K} \mathbb{E}[\mathbb{1}\{s = s_t^k\} g_t^{k-1}(s) + \mathbb{1}\{s \neq s_t^k\} g_t^{k-1}(s) \mid \mathcal{F}_{k-1}]$$

$$\qquad - \mathbb{E}[\mathbb{1}\{s = s_t^k\} f_t^{k-1}(s) + \mathbb{1}\{s \neq s_t^k\} g_t^{k-1}(s) \mid \mathcal{F}_{k-1}]$$

$$\overset{(3)}{=} \sum_{t=1}^{H} \sum_{s \in \tilde{\mathcal{S}}} \sum_{k=1}^{K} g_t^{k-1}(s) - \mathbb{E}[\mathbb{1}\{s = s_t^k\} f_t^{k-1}(s) + \mathbb{1}\{s \neq s_t^k\} g_t^{k-1}(s) \mid \mathcal{F}_{k-1}]$$

$$\overset{(4)}{=} \sum_{t=1}^{H} \sum_{s \in \tilde{\mathcal{S}}} \sum_{k=1}^{K} g_t^{k-1}(s) - \mathbb{E}[g_t^k(s) \mid \mathcal{F}_{k-1}]. \tag{40}$$

Relation (1) holds since for $s = s_t^k$ the vector $g_k^t$ is updated according by $f^{k-1}$. Relation (2) holds by adding and subtracting $\mathbb{1}\{s \neq s_t^k\} g_t^{k-1}(s)$ while using the linearity of expectation. (3) holds since for any event $\mathbb{1}\{A\} + \mathbb{1}\{A^c\} = 1$ and since $g_t^{k-1}$ is $\mathcal{F}_{k-1}$ measurable. (4) holds by the definition of the update rule,

$$\mathbb{E}[\mathbb{1}\{s = s_t^k\} f_t^{k-1}(s) + \mathbb{1}\{s \neq s_t^k\} g_t^{k-1}(s) \mid \mathcal{F}_{k-1}]$$
$$= \mathbb{E}[\mathbb{1}\{s = s_t^k\} \mid \mathcal{F}_{k-1}] f_t^{k-1}(s) + \mathbb{E}[\mathbb{1}\{s \neq s_t^k\} \mid \mathcal{F}_{k-1}] g_t^{k-1}(s)$$
$$= \Pr(s_t^k = s \mid \mathcal{F}_{k-1}) f_t^{k-1}(s) + \Pr(s_t^k \neq s \mid \mathcal{F}_{k-1}) g_t^{k-1}(s) = \mathbb{E}[g_t^k(s) \mid \mathcal{F}_{k-1}].$$

Where we used that $g_t^{k-1}(s)$ is $\mathcal{F}_{k-1}$ measurable and the assumption that $f_t^{k-1}(s)$ is $\mathcal{F}_{k-1}$ measurable in the first relation. □

The following lemma is a variant of a well known error propagation analysis in case of an approximate model.

**Lemma 16** (Model Error Propagation). *Let* $\|(P(\cdot \mid s, a) - \hat{P}(\cdot \mid s, a))\| \leq \epsilon_P$ *for any* $s, a$. *Then, for any policy* $\pi$,

$$\forall s_1 \in \mathcal{S}, \sum_{s_n} \left| P^\pi(s_n \mid s_1) - \hat{P}^\pi(s_n \mid s_1) \right| \leq n\epsilon_P$$

*Proof.* We prove the claim by induction. For the base case $n = 1$ we get that for any $s_1 \in \mathcal{S}$

$$\sum_{s_2} \left| P^\pi(s_2 \mid s_1) - \hat{P}^\pi(s_2 \mid s_1) \right|$$

$$= \sum_{s_2} \left| \sum_a \pi(a \mid s_1) \Big( P(s_2 \mid s_1, a) - \hat{P}^\pi(s_2 \mid s_1, a) \Big) \right|$$

$$\leq \sum_a \pi(a \mid s_1) \sum_{s_2} \left| P(s_2 \mid s_1, a) - \hat{P}^\pi(s_2 \mid s_1, a) \right|$$

$$= \sum_a \pi(a \mid s_1) \| P(\cdot \mid s_1, a) - P(\cdot \mid s_1, a) \|_1 \leq \epsilon_P.$$

Assume the induction step, i.e., assume the claim holds for $k = n - 1$. We now prove the induction step, i.e., for $k = n$

$$\sum_{s_n} \left| P^\pi(s_n \mid s_1) - \hat{P}^\pi(s_n \mid s_1) \right|$$

$$= \sum_{s_n} \left| \sum_{s_2} P^\pi(s_n \mid s_2) P^\pi(s_2 \mid s_1) - \hat{P}^\pi(s_n \mid s_2) \hat{P}^\pi(s_2 \mid s_1) \right|$$

$$\leq \sum_{s_n} \sum_{s_2} \left| P^\pi(s_n \mid s_2) P^\pi(s_2 \mid s_1) - \hat{P}^\pi(s_n \mid s_2) \hat{P}^\pi(s_2 \mid s_1) \right|$$

$$\leq \sum_{s_n} \sum_{s_2} \left| P^\pi(s_n \mid s_2) P^\pi(s_2 \mid s_1) - \hat{P}^\pi(s_n \mid s_2) P^\pi(s_2 \mid s_1) \right|$$

$$+ \left| \hat{P}^\pi(s_n \mid s_2) \hat{P}^\pi(s_2 \mid s_1) - \hat{P}^\pi(s_n \mid s_2) P^\pi(s_2 \mid s_1) \right|$$

$$\leq \sum_{s_n} \sum_{s_2} P^\pi(s_2 \mid s_1) \left| P^\pi(s_n \mid s_2) - \hat{P}^\pi(s_n \mid s_2) \right|$$

$$+ \hat{P}^\pi(s_n \mid s_2) \left| \hat{P}^\pi(s_2 \mid s_1) - P^\pi(s_2 \mid s_1) \right|$$

$$\leq \underbrace{\sum_{s_2} P^\pi(s_2 \mid s_1)}_{=1} \left( \max_{s_2'} \sum_{s_n} \left| P^\pi(s_n \mid s_2') - \hat{P}^\pi(s_n \mid s_2') \right| \right)$$

$$+ \sum_{s_2} \underbrace{\left( \sum_{s_n} \hat{P}^\pi(s_n \mid s_2) \right)}_{=1} \left| \hat{P}^\pi(s_2 \mid s_1) - P^\pi(s_2 \mid s_1) \right|$$

$$= \max_{s_2} \sum_{s_n} \left| P^\pi(s_n \mid s_2) - \hat{P}^\pi(s_n \mid s_2) \right| + \sum_{s_2} \left| \hat{P}^\pi(s_2 \mid s_1) - P^\pi(s_2 \mid s_1) \right|.$$

By the induction hypothesis and the base case,

$$\max_{s_2'} \sum_{s_n} \left| P^\pi(s_n \mid s_2') - \hat{P}^\pi(s_n \mid s_2') \right| \leq \epsilon(n - 1)$$

$$\sum_{s_2} \left| \hat{P}^\pi(s_2 \mid s_1) - P^\pi(s_2 \mid s_1) \right| \leq \epsilon_P,$$

from which we prove the induction step,

$$\forall s_1 \in \mathcal{S}, \ \|P^\pi(\cdot \mid s_1)_1 - \hat{P}^\pi(\cdot \mid s_1)\| = \sum_{s_n} \left| P^\pi(s_n \mid s_1) - \hat{P}^\pi(s_n \mid s_1) \right| \le n\epsilon_P.$$

$\square$

**Lemma 17.** *Let $V_t^*(s), \hat{V}_t^*(s)$ be the optimal values on the MDP $\mathcal{M}, \hat{\mathcal{M}}$, respectively, and let $V_t^\pi(s), \hat{V}_t^\pi(s)$ be the value of a fixed policy $\pi$ on the MDP $\mathcal{M}, \hat{\mathcal{M}}$, respectively. Then,*

$$i) \ \|V_1^* - \hat{V}_1^*\|_\infty \le \frac{H(H-1)}{2}\epsilon_P,$$

$$ii) \ \forall \pi, \|V_1^\pi - \hat{V}_1^\pi\|_\infty \le \frac{H(H-1)}{2}\epsilon_P.$$

*Proof.* Both claims follow standard techniques based on the Simulation Lemma [20, 30].

*(i)* Let $\Delta_t(s) := \hat{V}_t^*(s) - V_t^*(s), \Delta_t = \max_s |\Delta_t(s)|$. For $t = H$ we have that for all $s$

$$\Delta_H(s) = \max_a r(s,a) + \sum_{s'} \hat{P}(s' \mid s,a)\hat{V}_{H+1}^*(s') - \max_a r(s,a) + \sum_{s'} P(s' \mid s,a)V_{H+1}^*(s')$$

$$= \max_a r(s,a) + \sum_{s'} \hat{P}(s' \mid s,a) \cdot 0 - \max_a r(s,a) + \sum_{s'} P(s' \mid s,a) \cdot 0 = 0, \qquad (41)$$

and the base case holds. Assume the claim holds for any $t \ge k+1$, we now prove it holds for $t = k$. The following relations hold for any $s$,

$$\Delta_t(s) = \max_a r(s,a) + \sum_{s'} \hat{P}(s' \mid s,a)\hat{V}_{t+1}^*(s') - \max_a r(s,a) + \sum_{s'} P(s' \mid s,a)v_{t+1}^*(s')$$

$$\le r(s,a^*) + \sum_{s'} \hat{P}(s' \mid s,a^*)\hat{V}_{t+1}^*(s') - r(s,a^*) + \sum_{s'} P(s' \mid s,a^*)V_{t+1}^*(s')$$

$$= \sum_{s'} \hat{P}(s' \mid s,a^*)\hat{V}_{t+1}^*(s') - P(s' \mid s,a^*)V_{t+1}^*(s')$$

$$\le \sum_{s'} \hat{P}(s' \mid s,a^*) \underbrace{\left| \hat{V}_{t+1}^*(s') - V_{t+1}^*(s') \right|}_{:=\Delta_{t+1}(s')} + \left| P(s' \mid s,a^*) - \hat{P}(s' \mid s,a^*) \right| V_{t+1}^*(s')$$

$$\le \sum_{s'} \hat{P}(s' \mid s,a^*)|\Delta_{t+1}(s')| + (H-t)\epsilon_P$$

$$\le \Delta_{t+1} \sum_{s'} \hat{P}(s' \mid s,a^*) + (H-t)\epsilon_P = \Delta_{t+1} + (H-t)\epsilon_P$$

The second relation holds by choosing $a^*$ to maximize the first term first. The forth relation by adding and subtracting $\hat{P}(s' \mid s,a^*)\hat{V}_{t+1}^*(s')$ and standard inequalities. The fifth relation by the fact $V_{t+1}^*(s) \le H-t$ and the assumption that for all $s,a \ \|P(\cdot \mid s,a) - \hat{P}(\cdot \mid s,a)\| \le \epsilon_P$. The sixth by the fact $\hat{P}(\cdot \mid s,a)$ is a probability distribution and thus sums to 1.

Lower bounding $\Delta_t(s)$ using similar technique with opposite inequalities yields,

$$\Delta_t(s) \ge -(\Delta_{t+1} + (H-t)\epsilon_P),$$

and thus,

$$|\Delta_t(s)| \le \Delta_{t+1} + (H-t)\epsilon_P.$$

As the above holds for any $s$ it holds for the maximizer. Thus,

$$\Delta_t \le \Delta_{t+1} + (H-t)\epsilon_P.$$

Iterating on this relation while using $\Delta_H(s) = 0$ by (41),

$$\|V_1^* - \hat{V}_1^*\|_\infty = \Delta_1 \le \sum_{t=1}^{H}(H-t)\epsilon_P = \epsilon_P \sum_{t=1}^{H-1} t = \frac{H(H-1)}{2}\epsilon_P.$$

*(ii)* The proof of the second claim follows the same proof of the first claim, without while replacing the max operator with the a fixed policy $\pi$. □

**Lemma 18** (Total Contribution of Approximate Model Errors)**.** *Let* $d_n := -\frac{1}{2}(h-1)h\epsilon_P + (H - n)h\epsilon_P$. *Then,*

$$\sum_{n=0}^{\frac{H}{h}-1} d_{1+nh} = \frac{1}{2}H(H-1)\epsilon_P.$$

*Proof.* The following relations hold.

$$\sum_{n=0}^{\frac{H}{h}-1} d_{1+nh} = -\frac{1}{2}H(h-1)\epsilon_P + \sum_{n=0}^{\frac{H}{h}-1}(H-1-nh)h\epsilon_P$$

$$= -\frac{1}{2}H(h-1)\epsilon_P + H(H-1)\epsilon_P - h^2\epsilon_P \sum_{n=0}^{\frac{H}{h}-1} n$$

$$= -\frac{1}{2}H(h-1)\epsilon_P + H(H-1)\epsilon_P - \frac{1}{2}h^2\epsilon_P(\frac{H-h}{h})\frac{H}{h}$$

$$= -\frac{1}{2}H(h-1)\epsilon_P + H(H-1)\epsilon_P - \frac{1}{2}H(H-h)\epsilon_P$$

$$= -\frac{1}{2}H(H-1)\epsilon_P + H(H-1)\epsilon_P = \frac{1}{2}H(H-1)\epsilon_P.$$

□

# 15 Approximate Dynamic Programming in Finite-Horizon MDPs

**Algorithm 9** (Exact) $h$-DP

init: $\forall s \in \mathcal{S},\ \forall n \in [\frac{H}{h}],\ V_{nh+1}(s) = H - nh$
**for** $n = \frac{H}{h} - 1, \frac{H}{h} - 2, \ldots, 1$ **do**
  **for** $s \in \mathcal{S}$ **do**
    $V_{nh+1}(s) = \left(T^h V_{(n+1)h+1}\right)(s)$
  **end for**
**end for**
**return:** $\{V_{nh+1}\}_{n=1}^{H/h}$

**Algorithm 10** $h$-DP with Approximate Model

init: $\forall s \in \mathcal{S},\ \forall n \in [\frac{H}{h}],\ V_{nh+1}(s) = H - nh$
**for** $n = \frac{H}{h} - 1, \frac{H}{h} - 2, \ldots, 1$ **do**
  **for** $s \in \mathcal{S}$ **do**
    $V_{nh+1}(s) = \left(\hat{T}^h V_{(n+1)h+1}\right)(s)$
  **end for**
**end for**
**return:** $\{V_{nh+1}\}_{n=1}^{H/h}$

In this section, we follow standard analysis [20, 30] and establish bounds on the performance of approximate DP algorithms which update by an $h$-step optimal Bellman operator (2). We abbreviate this class of algorithms by $h$-ADP. See that unlike previous analysis [20, 30], we focus on finite horizon MDPs, which is the setup in which $h$-RTDP is analyzed. The different approximation setting we analyze in this section corresponds to the ones anlayzed for $h$-RTDP: approximate model, approximate value update, and approximate state abstraction.

As a reminder and for the sake of completeness, we start by considering $h$-DP Algorithm 9, which is the exact, approximate-free, version of the following $h$-ADP algorithms. The algorithm uses backward induction and a $h$-step optimal Bellman operator $T^h$ by which it outputs the values $\{V_{nh+1}\}_{n=2}^{\frac{H}{h}}$. Notice that it holds $\{V_{nh+1}\}_{n=2}^{\frac{H}{h}} = \{V_{nh+1}^*\}_{n=2}^{\frac{H}{h}}$ by standard arguments on the Backward Induction algorithm. Furthermore, $T^h$ can be solved by Backward induction with the total computational complexity of $O(SAh)$ by using Backward Induction. Thus, the total computational complexity of $h$-DP is $O(SAH)$ similar to the one of standard DP, e.g., Backward Induction.

In terms of space complexity, $h$-DP stores in memory $O(S\frac{H}{h})$ value entries. Observe that an $h$-greedy policy (3) w.r.t. $\{V_{nh+1}\}_{n=2}^{\frac{H}{h}}$ is an optimal policy, as these values are the optimal values as previously observed. Ultimately, one would like using these values to act in the environment by the optimal policy. If one uses the Forward-Backward DP (Section 9) to calculate such an $h$-greedy policy, then an extra $O(hS_h)$ space should be used, which results in total $O(S\frac{H}{h} + hS_h)$ space complexity (as in $h$-RTDP) that decrease in $h$ if $S_h$ is not too big (see Remark 2). Furthermore, the computational complexity of such approach is $O(HhAS_hS_1)$ which increases in $h$.

In next sections, we consider approximate settings of $h$-DP and establish that an $h$-greedy policy (3) w.r.t. the output values $\{V_{nh+1}\}_{n=2}^{\frac{H}{h}}$ has an equivalent performance to the asymptotic policy by which $h$-RTDP acts.

## 15.1 $h$-ADP with an Approximate Model

In the case of an approximate model, we replace the Bellman operator $T$ used in $h$-DP with $\hat{T}$, the Bellman operator of the approximate model $\hat{p}$ instead the true one $p$ (we assume $r$ is exactly known, which correspond to the assumption made in Section 6.1). This results in Algorithm 10. Similarly to Section 6.1, we assume $\|\hat{p}(\cdot \mid s, a) - p(\cdot \mid s, a)\|_{TV} \leq \epsilon_P$, for all $(s, a) \in \mathcal{S} \times \mathcal{A}$. Furthermore, denote $\pi_P^*$ as the optimal policy of the approximate MDP.

Equivalently to $h$-DP, Algorithm 10 returns the optimal values of the *approximate model* (Algorithm 10 can be interpreted as exact $h$-DP applied on the approximate model). Thus, the $h$-greedy policy w.r.t. to the outputs of Algorithm 10 $\{V_{nh+1}\}_{n=2}^{\frac{H}{h}}$ is the optimal policy of the approximate MDP, $\pi_P^*$. The performance of $\pi_P^*$ is measured by relatively to the performance of the optimal policy, i.e., we wish to bound $\|V_1^* - V_1^{\pi_P^*}\|_\infty$. This term represents the performance gap between the optimal policy of the 'real' MDP to the performance of the optimal policy of the approximate MDP evaluated on the real MDP, and is bounded in the following proposition.

**Proposition 19.** *Assume for all $(s,a) \in \mathcal{S} \times \mathcal{A} : \|\hat{p}(\cdot \mid s,a) - p(\cdot \mid s,a)\|_{TV} \leq \epsilon_P$ and let $\pi_P^*$ be the optimal policy of the approximate MDP. Then,*

$$\|V_1^* - V_1^{\pi_P^*}\|_\infty \leq H(H-1)\epsilon_P.$$

*Proof.* Let $\hat{V}^{\pi_P^*}$ be the optimal value on the approximate MDP. By using the triangle inequality, the first and second claim of Lemma 17 we conclude the proof,

$$\|V_1^* - V_1^{\pi_P^*}\|_\infty \leq \|V_1^* - \hat{V}_1^{\pi_P^*}\|_\infty + \|\hat{V}_1^{\pi_P^*} - V_1^{\pi_P^*}\|_\infty \leq H(H-1)\epsilon_P.$$

$\square$

---

**Algorithm 11** $h$-DP with Approximate Value Updates

---

init: $\forall s \in \mathcal{S}, \ \forall n \in [\frac{H}{h}], \ V_{nh+1}(s) = H - nh$
**for** $n = \frac{H}{h} - 1, \frac{H}{h} - 2, \ldots, 1$ **do**
   **for** $s \in \mathcal{S}$ **do**
      $\bar{V}_t^k(s) = \epsilon_V(s) + (T^h V_{t+h})(s)$
   **end for**
**end for**
**return:** $\{V_{nh+1}\}_{n=1}^{H/h}$

---

**Algorithm 12** $h$-DP with Approximate State abstraction

---

init: $\forall s \in \mathcal{S}, \ \forall n \in [\frac{H}{h}], \ V_{nh+1}(s) = H - nh$
**for** $n = \frac{H}{h} - 1, \frac{H}{h} - 2, \ldots, 1$ **do**
   **for** $s \in \mathcal{S}$ **do**
      $\bar{V}_{nh+1}(\phi(s)) = \min\{(T^h V_{(n+1)h+1})(s), \bar{V}_{nh+1}^k(\phi(s))\}$
   **end for**
**end for**
**return:** $\{V_{nh+1}\}_{n=1}^{H/h}$

---

### 15.2 $h$-DP with Approximate Value Updates

In the case of a approximate value updates Algorithm 9 is replaced by an value updates with added noise $\epsilon_V(s)$, by which Algorithm 11 is formulated. Similarly to the assumption used for $h$-RTDP with approximate value updates (see Section 6.2) we assume for all $s \in \mathcal{S}, |\epsilon_V(s)| \leq \epsilon_V > 0$. The following proposition bounds the performance of an $h$-greedy policy w.r.t. the values output by Algorithm 11.

**Proposition 20.** *Assume for all $s \in \mathcal{S}, |\epsilon_V(s)| \leq \epsilon_V$. Let $\pi_V^*$ be the $h$-greedy policy (3) w.r.t. output Algorithm 11. Then,*

$$\|V_1^* - V_1^{\pi_V^*}\|_\infty \leq \frac{2H}{h}\epsilon_V.$$

*Proof.* Let $\left\{\hat{V}_{nh+1}^*\right\}_{n=1}^{H/h}$ denote the output of Algorithm 11. We establish two claims which are of similarity to the two claims of Lemma 17. Combining the two we prove the result.

*(i)* The following relations hold for all $s \in \mathcal{S}$ and $n \in \{0\} \cup [\frac{H}{h} - 1]$.

$$\Delta_{1+nh}(s) := \hat{V}^*_{1+nh}(s) - V^*_{1+nh}(s)$$
$$= \epsilon_V(s) + T^h \hat{V}^*_{1+(n+1)h}(s) - T^h V^*_{1+(1+n)h}(s')$$
$$= \epsilon_V(s) + \max_{a_0,\dots,a_{h-1}} \mathbb{E}\left[\sum_{t'=0}^{h-1} r(s_{t'}, a_{t'}(s_{t'})) + \hat{V}^*_{t+h}(s_h) \mid s_0 = s\right] - \max_{a_0,\dots,a_{h-1}} \mathbb{E}\left[\sum_{t'=0}^{h-1} r(s_{t'}, a_{t'}(s_{t'})) + V^*_{t+h}(s_h) \mid s_0 = s\right]$$

$$(42)$$

The second relation holds by the updating equation and the third relation by definition (2). Let $\{\hat{a}_0, \hat{a}_1, .., \hat{a}_{h-1}\}$ be the set of policies maximizes the second terms, then, by plugging this sequence to the third term we necessarily decrease it. Thus,

$$(42) \leq \epsilon_V(s) + \mathbb{E}\left[\sum_{t'=0}^{h-1} r(s_{t'}, a_{t'}(s_{t'})) + \hat{V}^*_{t+h}(s_h) \mid s_0 = s, \{a_{t'}\}_{t'=0}^{h-1} = \{\hat{a}_{t'}\}_{t'=0}^{h-1}\right]$$
$$- \mathbb{E}\left[\sum_{t'=0}^{h-1} r(s_{t'}, a_{t'}(s_{t'})) + V^*_{t+h}(s_h) \mid s_0 = s, \{a_{t'}\}_{t'=0}^{h-1} = \{\hat{a}_{t'}\}_{t'=0}^{h-1}\right]$$
$$= \epsilon_V(s) + \mathbb{E}\left[\hat{V}^*_{t+h}(s_h) - V^*_{t+h}(s_h) \mid s_0 = s, \{a_{t'}\}_{t'=0}^{h-1} = \{\hat{a}_{t'}\}_{t'=0}^{h-1}\right]$$
$$= \epsilon_V(s) + \mathbb{E}\left[\Delta_{1+(n+1)h}(s) \mid s_0 = s, \{a_{t'}\}_{t'=0}^{h-1} = \{\hat{a}_{t'}\}_{t'=0}^{h-1}\right] \leq \epsilon_V + \|\Delta_{1+(n+1)h}\|_\infty.$$

The second relation holds by linearity of expectation, the third relation by definition, and the forth by assumption on $\epsilon_V(s)$ and by the standard bounded $E[X] \leq \|X\|_\infty$.

Repeating the above arguments while choosing the sequence which maximizes the third term in (42) allows us to lower bound (42) as follows

$$(42) \geq -\epsilon_V - \|\Delta_{1+(n+1)h}\|_\infty,$$

and thus,

$$\|\Delta_{1+nh}\|_\infty \leq \epsilon_V + \|\Delta_{1+(n+1)h}\|_\infty.$$

Solving the recursion while using $\Delta_{H+1}(s) = 0$ for all $s \in \mathcal{S}$ we get

$$\|\Delta_1\|_\infty \leq \frac{H}{h}\epsilon_V. \tag{43}$$

*(ii)* The following relations hold for all $s \in \mathcal{S}$ and $n \in [\frac{H}{h}]$.

$$\Delta^{\pi^*_V}_{1+nh}(s) := \hat{V}^*_{1+nh}(s) - V^{\pi^*_V}_{1+nh}(s)$$
$$= \epsilon_V(s) + \max_{a_0,..,a_{h-1}} \mathbb{E}\left[\sum_{t'=0}^{h-1} r(s_{t'}, a_{t'}(s_{t'})) + \hat{V}^*_{1+(n+1)h}(s_h) \mid s_0 = s\right]$$
$$- \mathbb{E}\left[\sum_{t'=0}^{h-1} r(s_{t'}, a_{t'}(s_{t'})) + V^{\pi^*_V}_{1+(n+1)h}(s_h) \mid s_0 = s, \pi^*_V\right].$$

By definition, the sequence which maximizes the second term is $\pi^*_V$ as it is the $h$-greedy policy w.r.t. $\hat{V}^*$. Using the linearity of expectation we get

$$\Delta^{\pi^*_V}_{1+nh}(s) = \epsilon_V(s) + \mathbb{E}\left[\hat{V}^*_{1+(n+1)h}(s_h) - V^{\pi^*_V}_{1+(n+1)h}(s_h) \mid s_0 = s, \pi^*_V\right]$$
$$= \epsilon_V(s) + \mathbb{E}\left[\Delta^{\pi^*_V}_{1+(n+1)h}(s_{1+(n+1)h}) \mid s_0 = s, \pi^*_V\right].$$

As for all $s$, $|\epsilon_V(s)| \leq \epsilon_V$, using the triangle inequality and $E[X] \leq \|X\|_\infty$ we get the following recursion,

$$\|\Delta^{\pi^*_V}_{1+nh}\|_\infty \leq \epsilon_V + \|\Delta^{\pi^*_V}_{1+(n+1)h}\|_\infty.$$

Using $\|\Delta_{1+H}^{\pi_V^*}\|_\infty = 0$ we arrive to its solution,

$$\|\Delta_1^{\pi_V^*}\|_\infty \le \frac{H}{h}\epsilon_V. \tag{44}$$

which proves the second needed result.

Finally, using the triangle inequality and the two proven claims, (43) and (44), we conclude the proof.

$$\|V_1^* - V_1^{\pi_V^*}\|_\infty \le \|V_1^* - \hat{V}_1\|_\infty + \|\hat{V}_1^* - V_1^{\pi_V^*}\|_\infty = \|\Delta_1\|_\infty + \|\Delta_1^{\pi_V^*}\|_\infty \le 2\frac{H}{h}\epsilon_V.$$

$\square$

### 15.3  $h$-**DP with Approximate State Abstraction**

When an approximate state abstraction $\{\phi_{1+nh}\}_{n=0}^{\frac{H}{h}-1}$ is given, Algorithm 9 can be replaced by an exact value update in the reduced state space $\mathcal{S}_\phi$, as given in Algorithm 12. This corresponds to updating a value $V \in \mathbb{R}^{S_\phi}$, instead a value $\mathbb{R}^S$. An obvious advantage of such an algorithm, relatively to $h$-DP, is its reduced space complexity, as it only needs to store $O(\frac{H}{h}S_\phi)$ value entries, instead of $O(\frac{H}{h}S)$ as $h$-DP.

Yet, as seen in Algorithm 12, its computational complexity remains $O(SAH)$ as it needs to uniformly update on the entire (non-abstracted) state space. Would have we being given a representative from each equivalence classes under $\phi_{1+nh}$ for every $n \in \{0\} \cup [\frac{H}{h}]^8$ we could suggest an alternative Backward Induction algorithm with computational complexity of $O(S_\phi AH)$. However, as we do not assume access to this knowledge, we are obliged to scan the entire state space, without further assumptions.

The following proposition bounds the performance of an $h$-greedy policy w.r.t. the values output by Algorithm 12. Similarly to the analysis of the corresponding $h$-RTDP algorithm (see Section 6.3), we assume $\{\phi_{1+nh}\}_{n=0}^{\frac{H}{h}-1}$ satisfy Assumption 1.

**Proposition 21.** *Let $\{\phi_{1+nh}\}_{n=0}^{\frac{H}{h}-1}$ satisfy Assumption 1. Let $\left\{\hat{V}_{nh+1}^*\right\}_{n=1}^{\frac{H}{h}}$ denote the output of Algorithm 12 and let $\pi_A^*$ be the $h$-greedy policy w.r.t. these approximate values* (3). *Then,*

$$\|V_1^* - V_1^{\pi_A^*}\|_\infty \le \frac{H}{h}\epsilon_A.$$

*Proof.* We establish two claims which are of similarily to the two claims of Lemma 17 and Proposition 20. Combining the two we prove the result.

*(i)* The following relations hold for any $s \in \mathcal{S}$.

$$
\begin{aligned}
&\hat{V}_{1+nh}^*(\phi_{1+nh}(s)) - V_{1+nh}^*(s) \\
&= T^h \hat{V}_{\phi,1+(n+1)h}^*(s) - T^h V_{1+(1+n)h}^*(s) \\
&= \max_{a_0,..,a_{h-1}} \mathbb{E}\left[\sum_{t'=0}^{h-1} r(s_{t'}, a_{t'}(s_{t'})) + \hat{V}_{1+(n+1)h}^*(\phi_{1+(n+1)h}(s_h)) \mid s_0 = s\right] \\
&\quad - \max_{a_0,..,a_{h-1}} \mathbb{E}\left[\sum_{t'=0}^{h-1} r(s_{t'}, a_{t'}(s_{t'})) + V_{1+(n+1)h}^*(s_h) \mid s_0 = s\right]
\end{aligned}
\tag{45}
$$

The second and third relation holds by the updating rule of Algorithm 12. Let $\{\hat{a}_0, \hat{a}_1, .., \hat{a}_{h-1}\}$ be the set of policies which maximizes the first term. Then, by plugging this sequence to the second

term we necessarily decrease it, and the following holds.

$$(45) \leq \mathbb{E}\left[\sum_{t'=0}^{h-1} r(s_{t'}, a_{t'}(s_{t'})) + \hat{V}^*_{1+(n+1)h}(\phi_{1+(n+1)h}(s_h)) \mid s_0 = s, \{a_{t'}\}_{t'=0}^{h-1} = \{\hat{a}_{t'}\}_{t'=0}^{h-1}\right]$$

$$- \mathbb{E}\left[\sum_{t'=0}^{h-1} r(s_{t'}, a_{t'}(s_{t'})) + V^*_{1+(n+1)h}(s_h) \mid s_0 = s, \{a_{t'}\}_{t'=0}^{h-1} = \{\hat{a}_{t'}\}_{t'=0}^{h-1}\right]$$

$$= \mathbb{E}\left[\hat{V}^*_{1+(n+1)h}(\phi_{1+(n+1)h}(s_h)) - V^*_{1+(n+1)h}(s_h) \mid s_0 = s, \{a_{t'}\}_{t'=0}^{h-1} = \{\hat{a}_{t'}\}_{t'=0}^{h-1}\right] \qquad (46)$$

Where the second relation holds by linearity of expectation. By Assumption 1 the following inequality holds,

$$(46) \leq \mathbb{E}\left[\hat{V}^*_{1+(n+1)h}(\phi_{1+(n+1)h}(s_h)) - \max_{\bar{s}_h \in \Phi_{1+(n+1)h}(s_h)} V^*_{1+(n+1)h}(\bar{s}_h) + \epsilon_A \mid s_0 = s, \{a_{t'}\}_{t'=0}^{h-1} = \{\hat{a}_{t'}\}_{t'=0}^{h-1}\right]$$

$$= \epsilon_A + \mathbb{E}\left[\hat{V}^*_{1+(n+1)h}(\phi_{1+(n+1)h}(s_h)) - \max_{\bar{s}_h \in \Phi_{1+(n+1)h}(s_h)} V^*_{1+(n+1)h}(\bar{s}_h) \mid s_0 = s, \{a_{t'}\}_{t'=0}^{h-1} = \{\hat{a}_{t'}\}_{t'=0}^{h-1}\right]$$

$$\leq \epsilon_A + \max_s \left|\hat{V}^*_{1+(n+1)h}(\phi_{1+(n+1)h}(s)) - \max_{\bar{s} \in \Phi_{1+(n+1)h}(s)} V^*_{1+(n+1)h}(\bar{s})\right| \qquad (47)$$

By choosing the sequence of polices which maximizes the second term in (45) and repeating similar arguments to the above we arrive to the following relations.

$$(45) \geq \mathbb{E}\left[\hat{V}^*_{1+(n+1)h}(\phi_{1+(n+1)h}(s_h)) - V^*_{1+(n+1)h}(s_h) \mid s_0 = s, \{a_{t'}\}_{t'=0}^{h-1} = \{\hat{a}_{t'}\}_{t'=0}^{h-1}\right]$$

$$\geq \mathbb{E}\left[\hat{V}^*_{1+(n+1)h}(\phi_{1+(n+1)h}(s_h)) - \max_{\bar{s}_h \in \Phi_{1+(n+1)h}(s_h)} V^*_{1+(n+1)h}(\bar{s}_h) \mid s_0 = s, \{a_{t'}\}_{t'=0}^{h-1} = \{\hat{a}_{t'}\}_{t'=0}^{h-1}\right]$$

$$\geq -\mathbb{E}\left[\left|\hat{V}^*_{1+(n+1)h}(\phi_{1+(n+1)h}(s_h)) - \max_{\bar{s}_h \in \phi^{-1}_{1+(n+1)h}(s_h)} V^*_{1+(n+1)h}(\bar{s}_h)\right| \mid s_0 = s, \{a_{t'}\}_{t'=0}^{h-1} = \{\hat{a}_{t'}\}_{t'=0}^{h-1}\right]$$

$$\geq -\max_s \left|\hat{V}^*_{1+(n+1)h}(\phi_{1+(n+1)h}(s)) - \max_{\bar{s} \in \Phi_{1+(n+1)h}(s)} V^*_{1+(n+1)h}(\bar{s})\right| \qquad (48)$$

Let $\Delta_{\phi,1+nh}(s) := \hat{V}^*_{1+nh}(\phi_{1+nh}(s)) - \max_{\bar{s} \in \Phi_{1+nh}(s)} V^*_{1+nh}(\bar{s})$. The following upper bound holds,

$$\Delta_{\phi,1+nh}(s) := \hat{V}^*_{1+nh}(\phi_{1+nh}(s)) - \max_{\bar{s} \in \phi^{-1}_{1+nh}(s)} V^*_{1+nh}(\bar{s})$$

$$\leq \hat{V}^*_{1+nh}(\phi_{1+nh}(s)) - V^*_{1+nh}(s)$$

$$\leq \max_s |\hat{V}^*_{1+(n+1)h}(\phi_{1+(n+1)h}(s)) - \max_{\bar{s} \in \Phi_{1+(n+1)h}(s)} V^*_{1+(n+1)h}(\bar{s})| + \epsilon_A$$

$$= \|\Delta_{\phi,1+(n+1)h}\|_\infty + \epsilon_A.$$

where the third relation is by (47). Furthermore, the following lower bounds holds,

$$\Delta_{\phi,1+nh}(s) := \hat{V}^*_{1+nh}(\phi_{1+nh}(s)) - \max_{\bar{s} \in \Phi_{1+nh}(s)} V^*_{1+nh}(\bar{s})$$

$$\geq \hat{V}^*_{1+nh}(\phi_{1+nh}(s)) - V^*_{1+nh}(s) - \epsilon_A$$

$$\geq -\max_s \left|\hat{V}^*_{1+(n+1)h}(\phi_{1+(n+1)h}(s)) - \max_{\bar{s} \in \Phi_{1+(n+1)h}(s)} V^*_{1+(n+1)h}(\bar{s})\right| - \epsilon_A$$

$$= -\|\Delta_{\phi,1+(n+1)h}\|_\infty - \epsilon_A,$$

where the second relation holds by Assumption 1 and the third by (48).

By the upper and lower bounds on $\Delta_{\phi,1+nh}(s)$ which holds for all $s$ we conclude that

$$\|\Delta_{\phi,1+nh}\|_\infty \leq \|\Delta_{\phi,1+(n+1)h}\|_\infty + \epsilon_A.$$

Using $\|\Delta_{\phi,H+1}\|_\infty = 0$ we solve the recursion and conclude that

$$\|\Delta_{\phi,1}\|_\infty \leq \frac{H}{h}\epsilon_A. \qquad (49)$$

*(ii)* The following relations hold based on similar arguments as in (42). Let $\Delta^{\pi_A^*}_{1+nh} := \max_s \hat{V}^*_{1+nh}(\phi_{1+nh}(s)) - V^{\pi_A^*}_{1+nh}(s)$. For all $s$ the following relations hold.

$$\hat{V}^*_{1+nh}(\phi(s)) - V^{\pi_A^*}_{1+nh}(s)$$

$$\leq \max_{a_0,..,a_{h-1}} \mathbb{E}\left[\sum_{t'=0}^{h-1} r(s_{t'}, a_{t'}(s_{t'})) + \hat{V}^*_{1+(n+1)h}(\phi(s_h)) \mid s_0 = s\right]$$

$$- \mathbb{E}^{\pi_A^*}\left[\sum_{t'=0}^{h-1} r(s_{t'}, a_{t'}(s_{t'})) + V^{\pi_A^*}_{1+(n+1)h}(s_h) \mid s_0 = s\right], \tag{50}$$

the first relation holds by the updating rule which update by the (see Algorithm 12), and since $V_t^\pi = (T^\pi)^h V_{t+h}^\pi$, similarly to the optimal Bellman operator (2).

By definition, the sequence which maximizes the first term is $\pi_A^*$ as it is the $h$-greedy policy w.r.t. $\hat{V}^*$. Using the linearity of expectation we get

$$(50) = \mathbb{E}^{\pi_A^*}\left[\hat{V}^*_{1+(n+1)h}(\phi(s_h)) - V^{\pi_A^*}_{1+(n+1)h}(s_h) \mid s_0 = s\right]$$

$$\leq \max_s \hat{V}^*_{1+(n+1)h}(\phi(s)) - V^{\pi_A^*}_{1+(n+1)h}(s) := \Delta^{\pi_A^*}_{1+(n+1)h}. \tag{51}$$

Since (51) for all $s$ it also holds for the maximum, i.e.,

$$\Delta^{\pi_A^*}_{1+nh} := \max_s \hat{V}^*_{1+nh}(\phi(s)) - V^{\pi_A^*}_{1+nh}(s) \leq \Delta^{\pi_A^*}_{1+(n+1)h}.$$

As $\Delta^{\pi_A^*}_{H+1} = 0$ and iterating on the above recursion we get,

$$\Delta^{\pi_A^*}_1 \leq 0. \tag{52}$$

We are now ready to prove the proposition. For any $s$ the following holds,

$$V_1^*(s) - V_1^{\pi_A^*}(s) = \underbrace{V_1^*(s) - \hat{V}_1(\phi(s))}_{(A)} + \underbrace{\hat{V}_1(\phi(s)) - V_1^{\pi_A^*}(s)}_{B}.$$

By (49)

$$(A) \leq \max_{\bar{s} \in \Phi_1(s)} V_1^*(\bar{s}) - \hat{V}_1(\phi_1(s))$$

$$:= -\Delta_{\phi,1}(s) \leq \|\Delta_{\phi,1}\|_\infty \leq \frac{H}{h}\epsilon_A.$$

By (52),

$$\hat{V}_1(\phi_1(s)) - V_1^{\pi_A^*}(s) \leq \max_{\bar{s}}\left(\hat{V}_1(\phi_1(\bar{s})) - V_1^{\pi_A^*}(\bar{s})\right) = \Delta^{\pi_A^*}_1 \leq 0.$$

Lastly, combining the above and using $V^* \geq V^\pi$, we get that for all $s$

$$0 \leq V_1^*(s) - V_1^{\pi_A^*}(s) \leq \frac{H}{h}\epsilon_A.$$

$$\rightarrow \|V_1^* - V_1^{\pi_A^*}\|_\infty \leq \frac{H}{h}\epsilon_A.$$

$\square$