[Reviews · NeurIPS 2020]

Review 1

Summary and Contributions: The paper analyzes the computational complexity and error sensitivity of RTDP with h-step-lookahead policy. RTDP is an asynchronous value iteration algorithm for planning in MDPs, and the paper shows that its sample complexity improves if h-step-lookahead policy is used, for h > 1. The paper also analyzes h-RTDP's performance in the presence of model errors, update errors, and state abstraction.

Strengths: The paper studies an interesting problem and puts an intuitive result about the connection between sample complexity of h-step RTDP and of its vanilla 1-step version on a firm theoretical footing. It also presents other interesting, less intuitive results on h-RTDP with approximate state abstraction.

Weaknesses: In my opinion, the paper has just one weakness, but a critical one. Qualitative result that using h-lookahead backups decreases sample complexity of RTDP is very much expected, because it essentially decreases the horizon of the problem by the factor of $h$. Note that in the limiting case of h=H, we would expect to need 1 sample to solve the problem. However, in this limiting case as well as less extreme ones, we aren't getting a free lunch: part of the computational cost that h-RTDP saves in sample complexity gets folded into computing the h-lookahead policies, i.e. running FB-DP. Thus, whether the paper's theory has bearing on reality is determined by the cost of computing h-lookahead policies, and this is where the paper is quite ambiguous. All it says on this (Section 3) is that if the number of states reachable by h-lookahead policies from a given state tends to be "small", the presented theory makes sense. There are two potential ways to address this: (1) provide a theoretical analysis characterizing the connection between _total_ computational complexity and h more precisely or (2) conduct an empirical evaluation. Unfortunately, the paper does neither. But for what values of h h-RTDP is worth it compared to RTDP and MCTS, and how sensitive h-RTDP is to the choice of h are questions without answering which the significance of the rest of the paper's theory is highly unclear.

Correctness: The theoretical claims are correct, I believe. However, an empirical methodology is missing entirely.

Clarity: Yes. There are just a few typos to fix: in many places the paper uses hyphens in expressions like "exhaustive-search", "time-steps", "backward-induction", etc, where none is needed.

Relation to Prior Work: Yes.

Reproducibility: Yes

Additional Feedback: ========================= COMMENTS AFTER REBUTTAL Thank you for your response. However, in this paper's case I find that the significance of the paper (i.e., support for your claim that "theoretical results provided in this work are important on their own") is severely lacking without experiments showing a link between this theory and an algorithm's performance in terms of measures like running time, number of 1-step Bellman backups, etc. ***Note: this is not a claim that every theoretical paper needs experiments; it applies only to this specific work, due to the theory issues mentioned in the original review.*** The rebuttal's attempted arguments against providing experiments really miss the mark: -- The rebuttal gives the "Beyond the one step greedy approach in RL" as an example of a paper similar in the degree of its theoretical focus to this submission, but that paper actually has experiments! -- The rebuttal says, "we feel a thorough empirical study of these algorithms is outside the scope of this work", but I never asked for a _thorough_ empirical study. Light experiments could do the job. That "Beyond the one step greedy approach in RL" paper that you mentioned yourself is a case in point. Its experiments are by no means thorough, having been done on toy gridworld, but they do help ground that paper's theory. Last but not least, the ICML-2020 version of this work was rejected precisely because 2 of 3 reviewers and the meta-reviewer thought the paper's significance was in doubt due to the lack of experiments. If the paper's theory is predictive of h-RTDP's actual performance, experiments showing this should be easy to do. Space limits aren't an issue -- such experiments don't need to be extensive, and there is plenty of material that can be moved to the appendix. And yet, the ICML reviewers' main criticism regarding significance has been left completely unaddressed in the NeurIPS version. The rebuttal's self-contradicting response and the repeated failure to address the issues pointed out during the ICML-2020 submission attempt make me doubt the significance of the paper's theoretical results even more. ========================= Providing a decent empirical evaluation along with the theory would help address my main concern. There is at least one recent work in a similar vein as this one: Manan Tomar, Yonathan Efroni, Mohammad Ghavamzadeh, "Multi-step Greedy Reinforcement Learning Algorithms", ICML-2020 that does exactly that: it also provides theory-based insights about h-lookahead versions of known algorithms, but backs them up with experiments. I believe this submission should do the same to analyze when its theory holds up. Benchmarks for such an evaluation are readily available from the International Probablilistic Planning Competitions -2006, 2008, 2011, and 2014, and an (L)RTDP implementation is available as part of the "mini-gpt" code project.


Review 2

Summary and Contributions: Real Time Dynamic Programming (RTDP) is a well-known algorithm for solving Markov Decision Processes (MDPs). In its standard form, it uses a 1-step lookahead policy to select actions during online planning. A h-step lookahead policy is also possible. When h>1, it is fair to assume that exploration will be better as well as the convergence to the optimal value function. The authors provide a rigorous theoretical justification that the increasing lookahead horizon leads to an improved sample complexity, which means that the algorithm converges faster using fewer interactions with the environment (either real or simulated in the case of RTDP). The result is rather expected, but the fact that the authors offer a solid theoretical ground to explain this intuitive result is important. The authors also investigate three approximate settings where the approximations are in the model, states or updates. This contribution seems to be important since it may lead to better alternatives to very widely-used MCTS methods.

Strengths: * The paper offers a strong theoretical contribution. * This work is timely since MCTS is one of the most popular AI techniques at the moment, and the paper provides important theoretical insights into alternatives. * The paper is very technical, yet writing is extremely accurate, rigours and clear.

Weaknesses: It is hard for me to find any limitations. Since the authors included some theoretical criticism of UCT in lines 52-29 (i.e. at the beginning of their paper), perhaps they could indicate at the end of their paper what their theoretical findings could mean in practice. In particular, do the results of this paper provide evidence that RTDP could be better than UCT on certain tasks? Which tasks would that be?

Correctness: I am very familiar with MDPs and reinforcement learning, but I don't prove PAC bounds in my work. This said I have to admit that I read this paper very carefully, and I did not notice any problems. All the derivations make an intuitive sense to me, and as I said above, the main result and the main point of this paper makes a lot of sense too.

Clarity: Clarity is excellent. One place where I was stuck was the beginning of section 5 (lines 147-166). Specifically, I did not see how this connects with "(t-1) mod h=0" in the pseudocode, but I can see that the authors tried to make this part clear. This sentence in line 207 should probably be revised "The case which exists error".

Relation to Prior Work: Prior work is clearly discussed. This paper would not be possible if the authors did not review relevant literature and did not know it. There is no problem at all here.

Reproducibility: Yes

Additional Feedback: I don't have anything to add. You did a great job. Thank you for your response. After reading it, I don't want to change my rating, and I believe that this is a good paper.


Review 3

Summary and Contributions: -- The paper proposes a multi-step real-rime dynamic programming (h-RTPD) algorithm for online planning in Markov decision processes (MDP). The authors theoretically analyze the performance of the algorithm in terms of regret and uniform-PAC criterion. They also provide the computational complexity of the algorithm, showing its effectiveness in comparison to conventional dynamic programming algorithms. ---- UPDATE ---- I want to thank the authors for their response to my comments. The paper has done a good job of theoretically analyzing an interesting question. However, given the concerns about the improvement of the aggregated computational burden, I find empirical evaluation a necessity. The main claim of the paper is improved complexity. In the proposed framework, there are two competing complexities: one from the number of planning and another from the cost per planning. So proving or demonstrating the improvement in the aggregated cost is crucial.

Strengths: -- Online planning plays a major role in designing efficient algorithms for control and learning in realistic problems. This paper provides theoretical support and sample complexity for a class of RTPD algorithms which can be interesting to the control and reinforcement learning communities. -- The authors also consider the proposed h-RTDP in the existence of three types of approximations, in the model, in the update of value function, and in the state abstraction, and extend their theoretical analysis to these settings. This will indeed make the algorithm more valuable in practical settings. -- The algorithmic contribution and theoretical claims of the paper seem sound.

Weaknesses: -- A drawback of the paper is lack of empirical evaluations. While the paper contains sufficient algorithmic and theoretical contributions, having empirical validation of the results will make the paper stronger. In particular, comparison with approximate dynamic programming (ADP) schemes and Monte Carlo tree search (MCTS) methods would be interesting. Furthermore, the readers would like to see the effect of look-ahead horizon on the empirical performance of the algorithm.

Correctness: -- The theoretical methodology and claims of the paper seem sound. -- There are no empirical results.

Clarity: -- The paper is well-written and is structurally clear to follow.

Relation to Prior Work: -- The authors adequately point to the relevant work.

Reproducibility: Yes

Additional Feedback: -- Please provide some intuition in the paper as to why the initial value is chosen optimistically, e.g., is it to induce exploration in the beginning? -- Some intuitive discussion as to why the h-RTDP algorithm improves sample complexity would help the understandability. Also, throughout the paper similar to the abstract, please emphasize that this improved sample complexity comes at the cost of additional computations. -- It would help to add the computational complexity of ADP and h-RTDP to Table 1. -- There are some inconsistencies in the bibliography. E.g., some first names are full and some are not, some words like “PAC” need capitalization, and “ICAPS” is fully written in [5] but only short version is provided in [6].


Review 4

Summary and Contributions: The paper analyzes an online planning algorithm h-RTDP, a h-step lookahead extension of RTDP. The primary contributions of the paper are to formally analyze the algorithmic benefits of using lookahead policies during online planning in different scenerios (exact, errors in the model, use of function approximation, etc.). This is an important area of study given the relatively weak guarantees yet strong empirical performance of methods like MCTS. Overall, the paper is extremely well written and contains a number of novel contributions. I think it would make an excellent addition to the online planning literature.

Strengths: Theoretical grounding, significance and novelty of the contribution, relevance to the NeurIPS community

Weaknesses: I wasn't able to identify any. An empirical evaluation is beyond the scope of this paper but would be nice to see in future work.

Correctness: Seems correct to me (but I didn't check the proofs carefully)

Clarity: Extremely well written.

Relation to Prior Work: Yes

Reproducibility: Yes

Additional Feedback: - The paper proposes and analyzes h-RTDP, a h-step lookahead online planning algorithm for H step problems. The main idea is to divide the full problem horizon H into smaller H / h intervals. The primary contribution of this paper is the excellent formal analysis of the tradeoffs introduced by the use of lookahead policies in horizon H online planning problems where a model of the environment is available or can be learned. Specifically, the paper relates the computational costs with regret bounds and sample complexity as a function of the planning horizon h and approximation errors, showing strong convergence guarantees. The paper is primarily analytical and doesn't include an empirical investigation. - The paper is extremely well-written. The rigorous analysis is well complemented with clear intuition. I only had a few minor comments (below). Overall, I think this paper would make an excellent addition to the online planning literature. Minor comments - Line 87: Should "tot" be on the RHS as well? $S_h^{tot} = max_s S_h^{tot} (s)$? - What happens if a state Does the definition of the cumulative (upto h) cardinality as the sum over h steps allow double counting of states reachable from s via paths of different lengths? - A few typos in the paper - Line 110: "instead of the naive approach" - Line 111: "FB-DP returns an action" - Line 124: "Proposition" - Line 267 "as a means to" - Line 316: "As less such values are used" is not entirely clear to me. UPDATE: I thank the authors for their detailed response. The other reviews have identified some potential areas of improvement. In particular, I share the common concern that the computational implications of h-step lookahead are currently quite unclear. I think this needs to be discussed in more detail in the context of how the proposed ideas could be leveraged in a practical online planning algorithm. I also encourage the authors to discuss the computational implications and tradeoffs of h-step lookahead compared to vanilla RTDP and consider doing some preliminary experiments to support the above discussion. I think these would strengthen the paper significantly. I've lowered my score a bit to reflect the above but I continue to be supportive of acceptance.

[Author Response · NeurIPS 2020]

We thank the reviewers for their comments.

We feel that the contributions of this work and the motivation for it are well understood by the reviewers: We investigate
the convergence properties of an online planning algorithm, given access to a lookahead policy oracle. Indeed, we
study the performance of such algorithms, both in the exact form and in several approximate settings, and compare
them to their approximate dynamic programming (ADP) counterparts. To the best of our knowledge, there is no
theoretical analysis regarding guarantees of lookahead policies in online planning algorithms. Furthermore, we believe
the generality of the presented techniques may be found useful in the analysis of and development of online planning
algorithms.

There is no dispute on the importance of empirical work. However, we believe that the theoretical results provided
in this work are important on their own. Our analysis spans not only the exact, but also three approximate settings,
and provide detailed comparison to the performance of ADP. Furthermore, unlike the scarcity of theoretical results in
online planning with lookahead policies, there are many works that study the empirical performance of different online
planning algorithms that are based on lookahead policy. Yet, the existing empirical works are heuristic; this stresses the
importance of theoretical results on online planning algorithms with lookahead policies. We hope the rigorous approach
pursued in our work will stimulate further theoretical research on the interplay between the lookahead horizon and the
performance of the online planning algorithm. There are, as always, interesting and important theoretical questions to
be solved.

As mentioned in response to Reviewer 3 and also clearly highlighted by Reviewer 4, a thorough empirical comparison
of RTDP, MCTS and ADP is very important and useful for the community, and may shed light on several unanswered
questions about the MCTS algorithm. However, due to the extent of our theoretical results and the length of current
paper, we feel a thorough empirical study of these algorithms is outside the scope of this work.

**R1.** For the comment on the needed empirical work, please see the above paragraphs that have been written for all
the reviewers. The second question that you raised is definitely of interest. In fact, in our opinion, answering such a
question deserves a work on its own, as there are probably several answers for it, such as which assumptions should be
made? What are the relevant structural properties? Can we choose it in an online manner? In a somewhat different
context, this question is equivalent to asking how to choose a hyperparameter of an algorithm? Indeed, it is an important
question, however it is outside the scope of the current work.

In the ICML-2020 paper "Multi-step Greedy Reinforcement Learning Algorithms" by Tomar, Efroni, and Ghavamzadeh,
the authors consider the framework of *model-free RL*, whereas we focus on online planning. For this reason, the
works are not very much related (we shall clarify this in the text, thanks!). Furthermore, this empirical paper is a follow
up to two theoretical papers ('Beyond the one step greedy approach in RL' ICML 18' and 'Multiple-Step Greedy
Policies in Approximate and Online RL' NeurIPS 19'). In our opinion, this clearly shows the importance of theoretical
results that can guide the design of practical algorithms.

**R2.** We would like to thank the reviewer for the supportive review. It definitely encourages us to keep investigating
online planning algorithms and expand our current understanding. We will revise the sentence on line 207.

Previous results which analyzed the performance of the UCT algorithm showed its worst-case sample complexity
depends exponentially (or even worst) on to the horizon of the problem. Furthermore, the sample complexity to find an
$\epsilon$ optimal action using the UCT usually scales as $O(1/\epsilon^2)$. Unlike the UCT, the sample complexity of RTDP depends
on the size of the state space (or the abstract state space as we showed in this work) and does not depend exponentially
on the horizon, only polynomially. The dependence on $\epsilon$ scales as $O(1/\epsilon)$. These results definitely implies on a possible
superiority of RTDP over MCTS from a theoretical perspective. We will emphasize it in the discussion part.

**R3.** We agree with the reviewer about the need for a thorough empirical work that compares RTDP, MCTS and
ADP. Such work is very important to the community in our opinion and might resolve the hardness of tuning the
hyper-parameters of the MCTS algorithm. Due to the extent of the theoretical results supplied in the paper, that
comprehensively study h-RTDP in its exact form and in three approximate settings, we feel a thorough empirical study
of these algorithms is outside the scope of the current paper.

We now address the additional feedback the reviewer gave. -) It is the same motivation as in RTDP. We will discuss this
in the final version of the paper. -) We tried to stress this as much as we can. We will add it to the abstract to avoid
confusions. -) Due to lack of space, we will add a table to the appendix to fully specify this computational complexity.
-) Thanks! we will fix this.

**R4.** We would like to thank the reviewer for the positive feedback. We also thank you for the minor comments which
helps us to improve the work. We will fix them all in the final version of the paper.

[Meta-Review · NeurIPS 2020]

This paper sparked lots of discussion, and an extreme spread in review scores. Concerns were raised about the significance of the results and the potential practical usefulness, especially because of concerns about the computational aspects of the algorithm, which were not deemed sufficiently addressed in either the paper or in the author response. For instance, the paper lacks numerical experiments (e.g., examples) that show in practice how such algorithms behave, and which could have aided the intuitive understanding, as well as avoid uncertainties. However, I want to give this paper the benefit of the doubt, mainly because of the inclusion in the results of analyses of the approximate cases, which expands the scope of settings for which the analysis is appropriate a great deal. Furthermore, I believe the topic of real-time planning to be an interesting direction for algorithm development, and that even if the current paper may not have proposed the best possible algorithm for that case I believe similar algorithms (e.g., with cheaper compute in finding suitable h-lookahead policies) could be developed by others in future work, inspired by - and learning from - this paper. In short, I hope the research community would benefit from being able to read and discuss this work, which is why I'm recommending to accept this paper.